# Mapping endothelial-cell diversity in cerebral cavernous malformations at single-cell resolution

**Fabrizio Orsenigo[1†], Lei Liu Conze[2†], Suvi Jauhiainen[2], Monica Corada[1], Francesca Lazzaroni[1], Matteo Malinverno[1], Veronica Sundell[2], Sara Isabel Cunha[2], Johan Brännström[2], Maria Ascención Globisch[2], Claudio Maderna[1], Maria Grazia Lampugnani[1,3]\*, Peetra Ulrica Magnusson[2]\*, Elisabetta Dejana[1,2]\***

[1]Vascular Biology Unit, FIRC Institute of Molecular Oncology Foundation (IFOM), Milan, Italy; [2]Department of Immunology, Genetics and Pathology, Uppsala University, Uppsala, Sweden; [3]Mario Negri Institute for Pharmacological Research, Milan, Italy

**Abstract** Cerebral cavernous malformation (CCM) is a rare neurovascular disease that is characterized by enlarged and irregular blood vessels that often lead to cerebral hemorrhage. Loss-of-function mutations to any of three genes results in CCM lesion formation; namely, *KRIT1*, *CCM2*, and *PDCD10 (CCM3)*. Here, we report for the first time in-depth single-cell RNA sequencing, combined with spatial transcriptomics and immunohistochemistry, to comprehensively characterize subclasses of brain endothelial cells (ECs) under both normal conditions and after deletion of *Pdcd10* (*Ccm3*) in a mouse model of CCM. Integrated single-cell analysis identifies arterial ECs as refractory to CCM transformation. Conversely, a subset of angiogenic venous capillary ECs and respective resident endothelial progenitors appear to be at the origin of CCM lesions. These data are relevant for the understanding of the plasticity of the brain vascular system and provide novel insights into the molecular basis of CCM disease at the single cell level.

\*For correspondence:
mariagrazia.lampugnani@ifom.eu
(MGL);
peetra.magnusson@igp.uu.se
(PUM);
elisabetta.dejana@ifom.eu (ED)

†These authors contributed
equally to this work

Competing interest: See
page 29

Reviewing editor: Salim
Abdelilah-Seyfried, Potsdam
University, Germany

## Introduction

Endothelial cells (ECs) are a particularly heterogenous cell population, as they have distinct structural, phenotypic, and functional properties (*Kalucka et al., 2020*; *Vanlandewijck et al., 2018*). Recent studies have shown that ECs of arteries, veins, and the lymphatic system show high degrees of specialization in their responses to specific hemodynamic and functional requirements (*Augustin and Koh, 2017*). Compared to large-vessel endothelia, this adaptation of microvascular ECs is seen by their expression of highly specialized properties, in terms of their responses to the specific requirements of different organs. Thus, an improved understanding of the molecular basis of this EC heterogeneity will help toward the definition of organ function under physiological and pathological conditions.

The cells that form the microcirculation of the brain represent a typical example of highly specialized microvascular ECs. The main important functions of these vessels relate to the regulation of cerebral blood flow and oxygen delivery, and the supply of energy metabolites to the nerve cells. Dysfunction of the brain vascular system can cause major problems for brain connectivity, synaptic activity, and information processing (*Zhao et al., 2015*). The specialized function of the brain microcirculation requires coordinated and continuous cross-talk between ECs and the other cell types, to thus establish what is known as the 'neurovascular unit' (*Zhao et al., 2015*). ECs are surrounded and embraced by pericytes and they are in contact with the glia (i.e. astrocytes, oligodendrocytes, microglia) and neurons (*Armulik et al., 2010*). The ECs in these neurovascular units form the blood–brain

barrier (BBB), which strictly controls the entry of neurotoxic plasma components, circulating inflammatory cells, and pathogens into the brain tissue (*Iadecola, 2017*).

As well as their control of general vascular permeability, the cells of the BBB promote the passage of nutrients and other essential molecules from the blood into the brain through their expression of multiple highly specialized transport systems (*Zhao et al., 2015*). Thus, the integrity of the BBB is regulated through coordinated and continuous interactions of the ECs with the surrounding neural cells, pericytes, and fibroblasts, which form a favorable environment for the ECs to develop and express their specialized properties.

Several genetic pathologies that can cause alterations in the development and function of the BBB arise as a result of inactivation of specific genes in the cells of the neurovascular units. While these diseases are relatively rare, they can help us to identify the key mechanisms that underlie the formation of the BBB and the acquisition of its specialized properties. A typical example here is the genetic disease known as cerebral cavernous malformation (CCM) (*Clatterbuck, 2001*). CCM vascular lesions, or cavernomas, mostly develop in the central nervous system and the retina. These can frequently bleed, which leads to epileptic seizures, focal neurological deficits, and other neurological problems, such as hemorrhagic stroke (*Clatterbuck, 2001*; *Labauge et al., 2007*; *Rigamonti et al., 1988*; *Wong et al., 2000*).

CCMs occur in both familial and sporadic forms. The familial form of CCM has an overall prevalence of less than 1:10,000. It is characterized by multiple CCM lesions that increase in number and size through the life of the patient, which results in recurrent cerebral hemorrhage. Familial CCM is due to loss-of-function mutations in any one of the three genes known as *KRIT1* (or *CCM1*), *CCM2* (or *OSM*), and *PDCD10* (or *CCM3*). These three types of CCM loss-of-function mutations show comparable morphology and specific brain localization. Instead, the sporadic form of CCM has relatively high prevalence, of about 1:200, and in the majority of cases, it occurs as a single cavernoma. Sporadic CCM can also result in a relatively variable set of symptoms, which are similar to familial CCM and depend on the location of the cavernoma (*Labauge et al., 2007*; *Akers et al., 2017*; *Cavalcanti et al., 2012*).

At present, to limit disease progression, there remains the need for an effective pharmacological treatment for patients with CCM, as to date, the only curative therapy is lesion eradication by surgical intervention or stereotactic radiosurgery. Open skull surgery is currently applied to selected symptomatic lesions only, as it is highly invasive, and it can result in significant complications, while being accompanied by unproven long-term benefit. Also, in familial CCM, neither surgery nor radiotherapy can cure multiple lesions throughout the brain and spinal cord. Therefore, despite many studies into CCM, an effective pharmacological therapy for this disease is still missing (*Abdelilah-Seyfried et al., 2020*).

An important step toward the understanding of CCM and any potential therapy is the identification of the EC populations that trigger and sustain the development of these vascular malformations. Here, we have used single-cell RNA sequencing (scRNA-seq) to map the transcriptional diversity of ECs in vascular cavernomas, with specific focus on *PDCD10*. Using this system, we identified distinct EC clusters and we define their functional roles in the development and progression of CCM.

## Results

### Endothelial cell heterogeneity across normal and CCM conditions

To study the EC subpopulation(s) that form lesions in CCM, we used *Cdh5*(PAC)-Cre-ER$^{T2}$/*Ccm3*$^{f/f}$/*Cldn5*(BAC)-GFP mice for EC-specific recombination (*Wang et al., 2010*). Following a single tamoxifen injection administered to the pups on their first day after birth, by P8 these mice develop CCM lesions in the brain that resemble human cavernomas (*Figure 1A–C*; *Figure 1—figure supplement 1*). ECs were isolated and enriched (CD45- and CD31+) from the brain of both *Cre*-negative (*Pdcd10*-wt) and *Cre*-positive (*Pdcd10*-ko) littermates and processed for droplet-based scRNA-seq. ScRNA-seq libraries were prepared using the 10× Genomics platform, and the R toolkit Seurat (version 3) (see Materials and methods) was used for integrated data analysis (*Figure 1A*).

After quality control, 15,057 cells from two *Pdcd10*-wt mice and 17,204 cells from two *Pdcd10*-ko mice were analyzed jointly. Through unsupervised clustering, 17 cell clusters (C) were distinguished and visualized using 'uniform manifold approximation and projection' (UMAP) (*Figure 1D–F*). Based

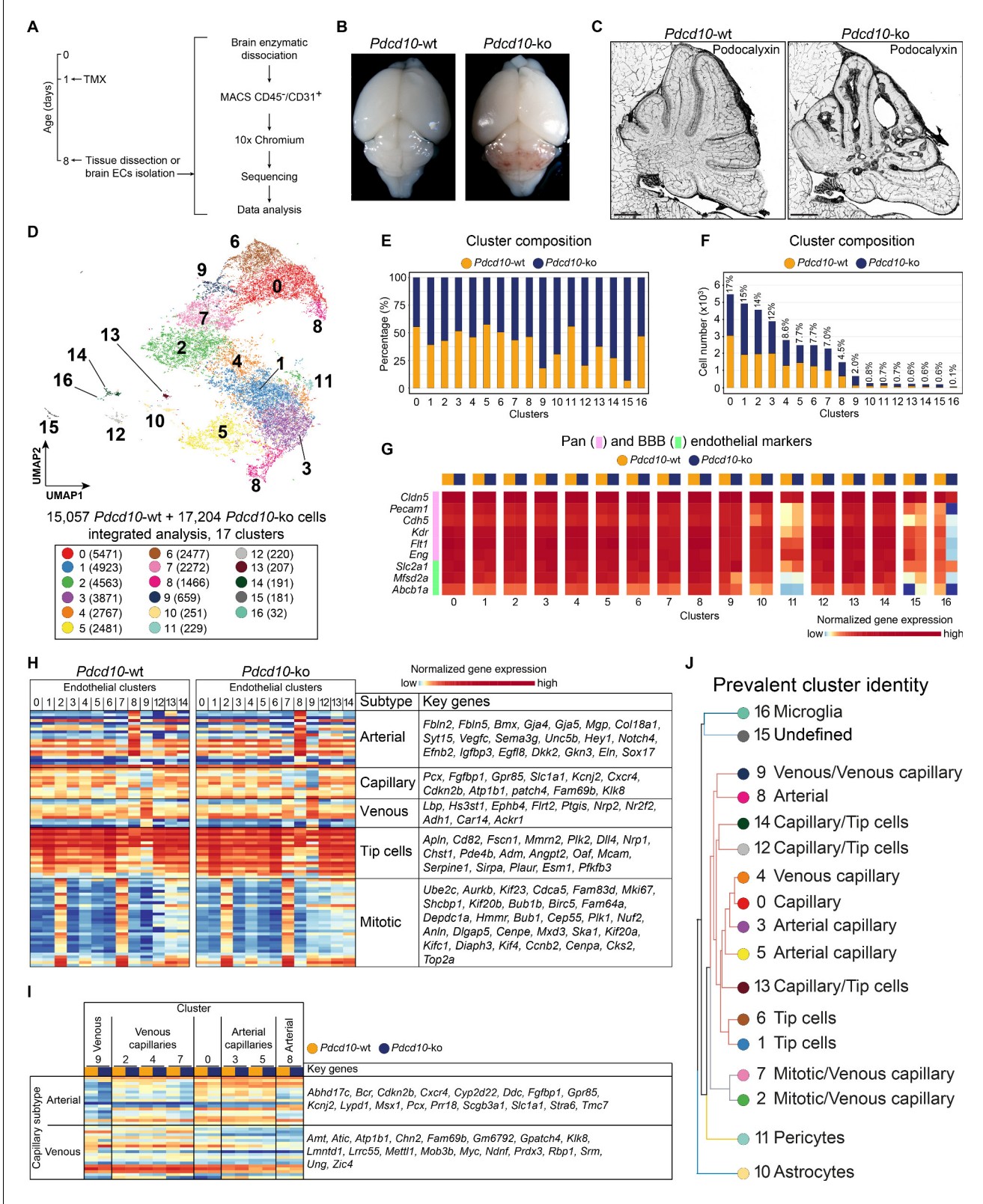

**Figure 1.** scRNA-sequencing of *Pdcd10*-wt and *Pdcd10*-ko endothelial cells. (**A**) Experimental scheme (see Materials and methods for details). (**B**) Representative photographs of *Pdcd10-wt* (left) and *Pdcd10-ko* (right) whole brains at P8. (**C**) Representative confocal microscopy of the vasculature of *Pdcd10-wt* (left) and *Pdcd10-ko* (right) cerebella at P8, stained for Podocalyxin (black; see also ***Figure 1—figure supplement 1***). Scale bars: 1 mm. (**D**) UMAP plot showing detected cell subpopulations in the *Pdcd10-wt* and *Pdcd10-ko* integrated analysis. The total numbers of cells within each cluster

*Figure 1 continued on next page*

*Figure 1 continued*

are shown in brackets in the color legend (bottom panel). (E) Plot of the percentages of *Pdcd10*-wt (orange) and *Pdcd10*-ko (blue) cells in each of the cluster. (F) Plot of the numbers of *Pdcd10*-wt (orange) and *Pdcd10*-ko (blue) cells in each cluster. The percentages of total cells in each cluster (%) is reported above each bar. (G) Heatmap of the selected pan- (pink) and blood–brain barrier- (BBB; green) endothelial cell markers (normalized expression shown; see Materials and methods) for the *Pdcd10*-wt and *Pdcd10*-ko cells in each cluster. (H) Heatmap of the selected endothelial cell subtype markers (from top to bottom, as indicated: arterial, capillary, venous, tip cells, mitotic), to show the normalized expression levels of the *Pdcd10*-wt (left) and *Pdcd10*-ko (right) cells in each cluster (see also ***Supplementary file 1***). For each subtype, the key genes are listed accordingly (top-to-bottom). (I) Heatmap of normalized expression of arterial and venous capillary markers in the capillary and mitotic/capillary clusters (C0, C2, C3, C4, C5, C7). The venous (C9; left) and arterial (C8; right) clusters are reported for reference. The *Pdcd10*-wt (orange) and *Pdcd10*-ko cells (blue) are shown separately. Key genes are listed according to the top-to-bottom order in the heatmap. (J) Summary of the prevalent identities of the 17 clusters based on endothelial cell subtype marker expression as in (H, I) and ***Figure 1—figure supplement 1***. The dendrogram of the hierarchical clustering is shown on the left. Non/mixed-ECs clusters (C10, C11, C15, C16) show early segregation. Arterial/venous (C8, C9), capillary (C0, C3, C4, C5), tip (C1 and C6) and mitotic/capillary (C2, C7) cells segregate as distinct groups of the final branches. Clusters 12, 13, and 14 show features of both capillaries and tip cells, but only C12 and C14 segregate on neighbor final branches.

The online version of this article includes the following figure supplement(s) for figure 1:

**Figure supplement 1.** Representative confocal microscopy of *Pdcd10*-wt and *Pdcd10*-ko brains at P8 immunostained for Podocalyxin.
**Figure supplement 2.** Expression of selected markers of endothelial and contaminant cells in the identified clusters.
**Figure supplement 3.** Sequencing quality control and clustering tree of examined *Pdcd10*-wt and *Pdcd10*-ko ECs.
**Figure supplement 4.** Clustering tree of examined *Pdcd10*-wt and *Pdcd10*-ko endothelial cells with overlaid prevalent cluster identity.

on the expression of a panel of known brain endothelial marker genes (***Figure 1G***; ***Figure 1—figure supplement 2A,B***), four clusters were identified as mixed with non-ECs (C10, C11, C15, C16). While the identity of C15 remains unclear, two clusters were identified as enriched in pericytes (C11) and microglia (C16) (***Figure 1—figure supplement 2C***; ***Supplementary file 1***). Cluster 10 (C10), instead, appeared to be a mixture of ECs and astrocytes, as marker genes of both of these cell types were highly expressed in this population (***Figure 1—figure supplement 2C***; ***Supplementary file 1***).

The remaining 13 endothelial subpopulations were then characterized on the basis of the overall expression levels of the reported endothelial marker genes (***Figure 1H,I***; ***Supplementary file 1***; ***Figure 1—figure supplement 2E***). These were categorized into six groups of ECs, as venous (C9), arterial (C8), capillary (C0, C3, C4, C5), tip (C1, C6), mitotic/capillary (C2, C7), and tip/capillary (C12, C13, C14) cells (***Figure 1H,J***). The capillary ECs were further distinguished into venous (C2, C4, C7) and arterial (C3, C5) (***Figure 1I,J***; ***Sabbagh et al., 2018***).

In our mouse model, CCM lesion formation correlated with deletion of the *Pdcd10* gene. All of the endothelial subpopulations except C13 showed significant down-regulation of *Pdcd10* expression after *Pdcd10* deletion (***Figure 1—figure supplement 2D***). As expected, the mixed endothelial and non-EC clusters (C10, C11, C15, C16) did not show *Pdcd10* down-regulation. Therefore, these five clusters were excluded from further analysis (***Figure 1—figure supplement 2D***).

## Pdcd10 deletion induces specific transcriptional profiles in distinct endothelial subpopulations

To determine whether the formation of cavernomas occurs for specific subpopulations of ECs, we analyzed the changes in gene expression between the *Pdcd10*-wt and *Pdcd10*-ko ECs in each cluster. The comparisons between the total numbers of differentially expressed genes (DEGs) indicated that the venous/venous capillary ECs in C9 were highly modified by *Pdcd10* deletion, while the arterial ECs in C8 were essentially not affected (***Figure 2A***). Also, the arterial- capillary ECs of C3 and C5 were minimally affected by *Pdcd10* deletion (***Figure 2A***). Furthermore, this analysis also showed that the tip cell C1 and C6 and the mitotic/venous capillary C7 and capillary C0 *Pdcd10*-ko EC clusters showed strong modifications to their gene expression, which indicated that *Pdcd10* deletion was particularly significant in these cells, as compared to the other cell clusters.

## Lesion marker genes

The transcription factors *Klf4* and *Klf2* are key drivers of the *Pdcd10-ko* phenotype, and they are up-regulated in brain ECs after *Pdcd10* deletion (***Maddaluno et al., 2013***; ***Zhou et al., 2016***; ***Cuttano et al., 2016***). Here, both *Klf4* and *Klf2* were up-regulated in the *Pdcd10*-ko ECs compared to the *Pdcd10*-wt ECs in almost all of the clusters (C0–C7, C9). The exceptions here were the arterial

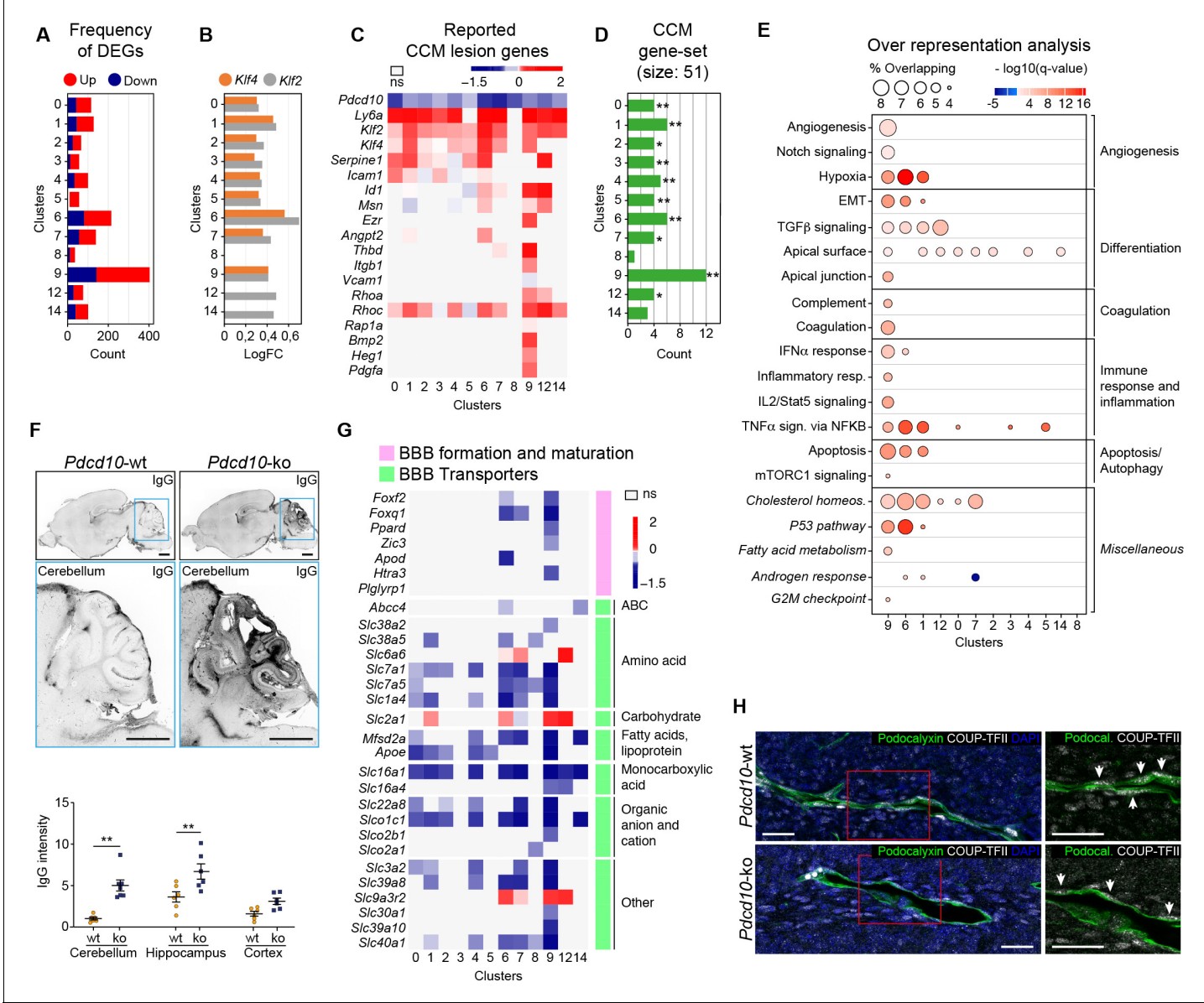

**Figure 2.** *Pdcd10* deletion induces specific transcriptional profiles in distinct endothelial cell subpopulations. (**A**) Numbers of significant differentially expressed genes (DEGs) (padj <0.05) in each cluster, showing up-regulation (red) and down-regulation (blue). (**B**) Average log fold changes of *Klf4* (orange) and *Klf2* (gray) expression in each cluster (*Pdcd10-ko* vs. *Pdcd10-wt*; padj <0.05). (**C**) Heatmap of selected known CCM lesion markers, as average logFC (padj <0.05) of *Pdcd10-ko versus Pdcd10-wt* cells (see also ***Supplementary file 2***). (**D**) Enrichment of CCM-associated genes (source: Rare Diseases GeneRIF Library) among DEGs (*Pdcd10-ko* vs. *Pdcd10-wt*) for each cluster (see Materials and methods). X-axis indicates the number of DEGs identified as CCM-associated genes in each cluster. Asterisks show the significance of enrichment: *p<0.05; **p<0.01. (**E**) Over-represented Molecular Signatures Databases hallmark gene sets in DEGs (average |logFC| ≥ 0.3; padj <0.05, see Materials and methods). The sizes of the dots reflect the proportion (%) of overlap between the DEGs and the reference gene sets, while the intensities of the colors show the -log10(q-value), color coded in red for up-regulated gene sets and in blue for down-regulated gene sets. Gene sets in italics have not been described previously for CCM. (**F**) Representative confocal microscopy of IgG leakage in *Pdcd10-wt* (left) and *Pdcd10-ko* (right) brain sections. Bottom images: higher magnification of the cerebellum (light blue boxed areas at top). Higher magnification of the hippocampus and cortex are shown in ***Figure 2—figure supplement 1***. Scale bars: 1 mm. Bottom panel: Quantification of IgG leakage (mean ± SEM; **p<0.01; ANOVA followed by Sidak multiple comparisons). *Pdcd10-wt*, n = 6; *Pdcd10-ko*, n = 7. (**G**) Heatmap of log fold expression changes of selected genes (padj <0.05) important for BBB formation and maturation (pink), and typical BBB transporters (green) between *Pdcd10-ko* and *Pdcd10-wt*. (**H**) Representative confocal microscopy of the venous marker COUP-TFII (encoded by the *Nr2f2* transcript, white), Podocalyxin (pan-endothelial, green) and DAPI (blue) of a *Pdcd10-wt* vessel (top) and a *Pdcd10-ko* lesion (bottom), both in the cerebellum. Arrows, COUP-TFII–positive endothelial nuclei. Scale bars: 25 μm.

The online version of this article includes the following source data and figure supplement(s) for figure 2:

**Source data 1.** Source data file for ***Figure 2F***.

*Figure 2 continued on next page*

*Figure 2 continued*

**Figure supplement 1.** *Pdcd10* deletion increases vessel permebility and impairs endothelial cell-to-cell junction organization.

C8 ECs, which were not affected (further commented below), and the tip/capillary C12 and C14 ECs, which showed significant up-regulation of only *Klf2* (*Figure 2B*). The greatest up-regulation of *Klf4* and *Klf2* was seen for the tip cell clusters C1 and C6. *Pdcd10*-wt arterial ECs of C8 showed significantly higher expression of *Klf4* and *Klf2* compared with *Pdcd10*-wt ECs of all the other clusters (average logFC, 0.72, 0.55, respectively; padj <0.05 for both), and there were no differences in *Klf4* or *Klf2* expression between the *Pdcd10*-wt and *Pdcd10*-ko ECs.

Next, we asked how the reported lesion-marker genes are regulated within each cluster from the *Pdcd10*-wt ECs to the *Pdcd10*-ko ECs (*Figure 2C*; *Supplementary file 2*, references). For the *Pdcd10*-ko, the venous/venous capillary C9 ECs showed up-regulation of most of the lesion-marker genes, followed by the mitotic/venous capillary C7 and tip cell C1 and C6 ECs. Moreover, we analyzed enrichment of CCM-related genes ('cerebral cavernous malformation' gene set from Rare Diseases GeneRIF library) among DEGs by using EnrichR (see Materials and methods). Here, the DEGs from cluster C9 showed the greatest overlap with the CCM gene set (12 of 51 CCM genes) (*Figure 2D*; *Supplementary file 3*). Once again, these observations strongly support the hypothesis that the venous/venous capillary C9 EC cluster is the strongest candidate cell cluster for contributions to the most advanced lesion development after *Pdcd10* deletion.

## CCM-related biological processes

To determine the functional impact of *Pdcd10* deletion in the different endothelial subpopulations, we analyzed the biological processes associated to the DEGs (as average |logFC| > 0.3) between the *Pdcd10*-wt and *Pdcd10*-ko using the Gene Set Enrichment Analysis (GSEA) software and the 'hallmark' gene sets from the Molecular Signatures Databases (version 7.0). In the absence of *Pdcd10* in the venous/venous capillary C9 and tip cell C1 and C6 ECs, the biological processes implicated in CCM pathology were mainly altered, which included angiogenesis, cell differentiation, coagulation, immune responses, and cell apoptosis (*Figure 2E*; *Supplementary file 4*). In a previous study, we reported that ECs undergo endothelial-to-mesenchymal transition that is mediated through increased TGFβ signaling (*Maddaluno et al., 2013*). This process is relevant for the development of vascular lesions, and is further confirmed here by this single-cell analysis (*Supplementary file 4*).

## State of BBB maturation and permeability

In patients with CCM, the BBB is impaired (*Clatterbuck, 2001*; *Mikati et al., 2015*), as reported here in this murine *Pdcd10*-depletion model, where there was significant increase in vascular permeability to endogenous immunoglobulins (*Figure 2F*, cerebellum; *Figure 2—figure supplement 1A*, cerebrum). Coherent with this, the expression levels of many regulators of BBB formation and maturation that cooperate to establish the BBB (e.g. *Foxf2*, *Foxq1*, *Ppard*, *Zic3*) (*Hupe et al., 2017*) were decreased in the potential lesion-forming C9 and C6 clusters (*Figure 2G*, pink). Transporters that finely tune BBB functions (*Figure 2G*, green) were also diffusely down-regulated in the *Pdcd10*-ko cells, with the C9, C6, and C7 ECs as the most affected. Of note, three transporters (i.e. *Slc2a1* or *Glut1*, *Slc9a3r2*, *Slc6a6*) were all up-regulated in the *Pdcd10*-ko cells of the C1, C6, C7, C9, and C12 ECs. Interestingly, enhancement of glucose transport mediated by *Slc2a1* supports glycolytic metabolism in ECs of tumor vessels (*Rohlenova et al., 2018*). All of these functions indicated were minimally or not modified in the arterial C8 *Pdcd10*-ko ECs (*Figure 2A–G*) and in the putative arterial-capillary C3 and C5 ECs (*Figure 1J*), which were only partially affected by the *Pdcd10* deletion (*Figure 2A–G*).

As previous reports have shown disrupted endothelial cell-to-cell junctions in CCM lesions (*Bravi et al., 2016*), we investigated the gene expression levels of other key junctional components. Among the panel of junctional molecules investigated (*Figure 2—figure supplement 1B*), only *Cldn5* expression was down-regulated in the C1, C6, and C9 ECs. Little or no changes were seen for the expression of other adherens and tight-junction components (*Figure 2—figure supplement 1B*). However, the tight-junction proteins Claudin-5, and Cingulin, as well as the adherens

junction VE-cadherin were not correctly localized at the cell-to-cell contacts in the ECs of the cavernomas of the *Pdcd10*-ko mice (*Figure 2—figure supplement 1C–E*).

In summary here, we have identified four endothelial subpopulations that are particularly affected by the deletion of the *Pdcd10* gene: the venous/venous capillary C9, tip cell C1 and C6, and mitotic/venous capillary C7 EC clusters. Among these, under the *Pdcd10*-ko conditions here, the venous/venous capillary C9 ECs represent the strongest candidate to form the most advanced lesions, or the most progressed lesion areas. The venous nature of the ECs in the cavernomas was further confirmed by expression of the venous marker COUP-TFII in vivo in the CCM lesions. *Figure 2H* shows the ECs in a large cavernoma that can be seen to be strongly positive for the venous marker COUP-TFII, which is encoded by *Nr2f2*.

## Arterial differentiation prevents ECs from forming CCM lesions

Despite the successful deletion of the *Pdcd10* gene (*Figure 1—figure supplement 2D*), gene expression of the arterial ECs (C8) in the *Pdcd10*-ko mice was almost unchanged from the effects of *Pdcd10* deletion. Indeed, in the comparison of the *Pdcd10*-wt and *Pdcd10*-ko mice, these arterial ECs showed the following: the lowest total number of DEGs (*Figure 2A*); no increase in *Klf4* and *Klf2* expression (*Figure 2B*); no, or minimal, signs of increased expression of lesion markers (*Figure 2C–E*); and no, or minimal, decreases in BBB markers and BBB transporters (*Figure 2G*).

Unique marker genes that are conserved between *Pdcd10*-wt and *Pdcd10*-ko cells specify the particularity of the transcriptome of each cluster (see Materials and methods). Therefore, to investigate the features of these arterial ECs that might promote their refractoriness to lesion formation, we identified the unique marker genes that were expressed in this C8 cluster (threshold: specificity average logFC >0.25; padj <0.05) (*Figure 3A*). Annotation of these genes according to Mouse Genome Informatics and PubMed (see Materials and methods) included homeostasis regulators (23 genes), oncosuppressors (12 genes), positive regulators of angiogenesis (nine genes), inhibitors of TGFβ signaling (seven genes), inflammation and response to stress (eight genes), inhibitors of angiogenesis (six genes), and positive regulators of BBB maturation and maintenance (six genes) (*Figure 3A*; *Supplementary file 5*, references). As several of these processes are indeed de-regulated in CCM malformations (*Clatterbuck, 2001*; *Labauge et al., 2007*; *Abdelilah-Seyfried et al., 2020*; *Maddaluno et al., 2013*; *Bravi et al., 2016*; *Bravi et al., 2015*), we speculate that the overexpression of at least some of these genes, alone or in concert, might contribute to maintenance of the normal phenotype of *Pdcd10*-deleted arterial ECs. Out of a total of 111 unique marker genes that were highly expressed in the C8 EC cluster, 80 had potential preventive functions, five had potential promotive functions, and 13 had potential dual functions, which depended on the cellular context (*Figure 3A,B*). These specific functions of the C8 cluster might also be supported by the group of four regulators of DNA remodeling (i.e. *Chd3*, *Gadd45g*, *Hey1*, *Mapk3*) and 10 transcriptional factors (*Gata2*, *Lmo7*, *Mecom*, *Msx1*, *Notch1*, *Notch4*, *Smad7*, *Sox13*, *Sox17*, *Sox18*) that were uniquely up-regulated in these arterial C8 ECs (*Figure 3D*). Notably, the levels of expression of *Klf2* and *Klf4* do not directly correlate with the resistance to *Pdcd10* deletion. As indicated above, *Klf2* and *Klf4* show the highest expression in arterial C8 ECs, with no difference between *Pdcd10*-wt and *Pdcd10*-ko. This suggests that high levels of *Klf2*/*Klf4* are not sufficient per se to trigger the *Pdcd10* phenotype.

To further understand the differences in the responses to the *Pdcd10* deletion between C8 and the other EC clusters, we annotated the few genes that were uniquely differentially expressed in the *Pdcd10*-ko ECs of the arterial C8 cluster only in comparison to *Pdcd10*-wt ECs of C8 (unique DEGs; six genes upregulated and one gene downregulated) (*Figure 3C*). Most of these have potential preventive roles against lesion formation, as they positively regulate processes that are dysfunctional in *Pdcd10*-deleted ECs and might contribute to the refractoriness of arterial *Pdcd10*-ko ECs to express a mutant phenotype. In particular, *Fbln2*, *Fbln5*, *Tmem100*, and *Nebl* cooperate in BBB protection, vascular differentiation, and maintenance of vascular integrity and mechano-sensation (*Supplementary file 6*, references). These four genes, and also *Stmn2*, are markers of arterial ECs in adult mice (*Vanlandewijck et al., 2018*). Furthermore, *Emp2* promotes HIF-1-alpha dependent vascular endothelial growth factor production and assembly of capillary-like tubes (*Supplementary file 6*).

Among the few uniquely differentially expressed genes between the *Pdcd10*-wt and the *Pdcd10*-ko in these arterial C8 ECs, *Slco2a1* was the only one that was down-regulated in the *Pdcd10*-ko.

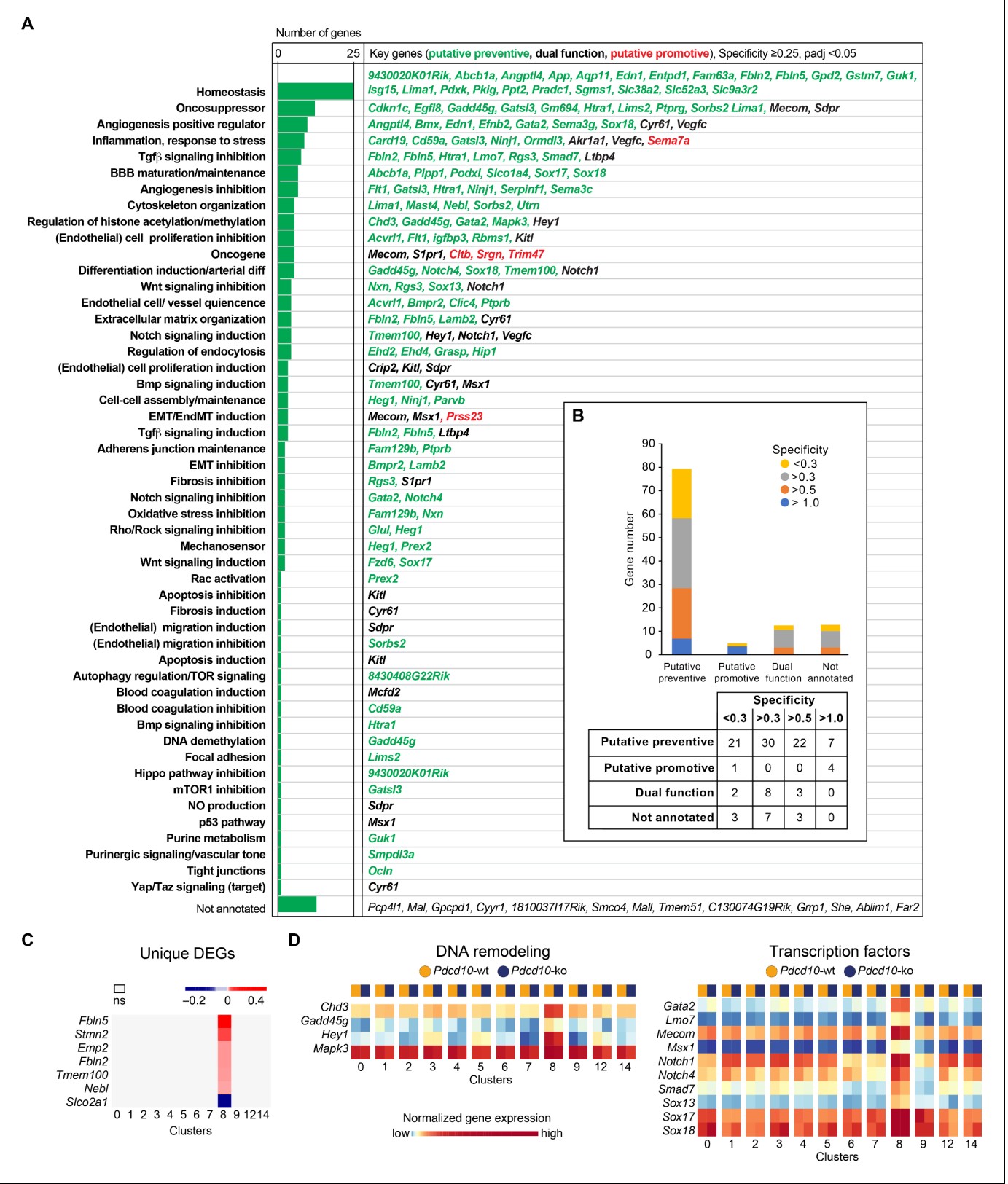

**Figure 3.** Unique functional markers suggest arterial *Pdcd10*-ko ECs may be protected from forming cavernomas. (**A**) Annotation of C8 unique marker genes. 111 unique marker genes were identified using a log fold change (specificity) cutoff of 0.25 and padj <0.05. Mouse Genome Informatics (http://www.informatics.jax.org/) and PubMed (https://pubmed.ncbi.nlm.nih.gov/) were used for manual annotation (keywords: 'gene name AND

*Figure 3 continued on next page*

Figure 3 continued

entothelial cells' or 'gene name' if the previous search gave zero results). Based on the functions of these genes and the deregulation of these functions reported in *Pdcd10*-ko lesions, we tentatively categorized these unique marker genes into 'preventive' (green), 'promotive' (red) and 'dual-function' (black) (See details of annotation in *Supplementary file 5* and *6*). Here, 13 unique marker genes either lack clearly defined functionality or have non-EC related functions and could not be annotated. (B) Statistics of annotated unique marker genes in C8. Each gene was counted only once, although it can appear in more than one category in (A). (C) Heatmap of log fold changes of uniquely differentially expressed genes (unique DEGs) in C8. Unique DEGs were identified using thresholds: average |logFC| > 0.2 and padj <0.05. (D) Heatmap of normalized expression levels of the selected C8 unique marker genes (left: regulators of DNA remodeling; right: transcription factors) in the *Pdcd10*-wt and *Pdcd10*-ko cells in each cluster.

These lower levels of this transporter might have preventive activity on the arterial ECs (*Supplementary file 6*).

Taken together, it appears that these arterial ECs selectively express genes that positively regulate processes that are altered after the *Pdcd10* deletion. We hypothesize that these genes can counteract the effects of the *Pdcd10* deletion, and thus prevent lesion formation.

It remains to be explored how microenvironmental cues (e.g. heterotypic cell-cell, cell-matrix interactions, type of shear stress *Li et al., 2019*) contribute to shape the arterial transcriptome and confer resistance to the effects of *Pdcd10* deletion.

## Cluster 9 Pdcd10-ko shares venous and angiogenic (mitotic and tip cell) traits

The number of *Pdcd10*-ko cells was significantly increased over the respective *Pdcd10*-wt cells for C9 (4.8-fold), C12 (4.0-fold), and C14 (2.8-fold) (*Figure 1E,F*; *Figure 4A*).

Cluster nine enrichment in the *Pdcd10*-ko cells is likely due to sustained proliferation, as indicated by the greater proportion of the *Pdcd10*-ko cells (vs. *Pdcd10*-wt) that expressed mitotic markers (*Figure 4B*). Furthermore, *Pdcd10*-ko cells of C9 expressed significantly higher levels of several mitotic markers that regulate different phases of mitosis (*Figure 4C*), including: G2/M (*Ccnb2*); mitotic spindle orientation (*Cenpa*); sister chromatid segregation (*Top2a*); and exit from mitosis (*Ube2s, Ube2c, Birc5*) (see references in *Supplementary file 1*).

In addition to showing mitotic features, the scRNA-seq data reported in previous sections supported the concept that the *Pdcd10*-ko ECs of C9 take part in the development of lesions. Accordingly, increased numbers of mitotic ECs (i.e. co-expression of Ki-67 and ERG *Birdsey et al., 2015*; *Booth and Earnshaw, 2017*) were seen for cavernomas, but not in the pseudo-normal vessels of the *Pdcd10-ko* brain (*Figure 4D*, representative immunofluorescence; *Figure 4E*, quantification; *Video 1*).

To validate the scRNA-seq data in situ, the Visium technology (also known as Spatial Transcriptomics) was applied. We first assessed the quality of the data obtained by manual annotation of the Visium clusters (see Materials and methods). Briefly, the diverse anatomical regions of the cerebellum were identified (*Figure 4—figure supplement 1A*; e.g. internal granular layer, red; external granular layer, orange; molecular layer, green). The spots that expressed pan-endothelial and lesion-endothelial markers were then identified (*Figure 4—figure supplement 1B*), among which those that co-expressed *Mki67* were counted. This analysis of the Visium data confirmed enhanced proliferation specifically in the ECs of the cavernomas (*Figure 4F,G*).

For the tip cell traits, as the other crucial phenotype in angiogenesis, increased proportions of the *Pdcd10*-ko cells expressed a set of tip cell markers in C9 (i.e. *Plaur*, *Apln*, *Mmrn2*) and C12 (i.e. *Plaur*) (*Figure 5A*). Moreover, the expression levels of some tip cell markers were also significantly increased in the *Pdcd10*-ko cells of several of the clusters. While tip cell clusters C1 and C6 showed the greatest number of up-regulated tip cell markers (*Figure 5B*), C9 *Pdcd10*-ko cells showed up-regulated expression of only a few tip cell markers (*Figure 5B*, *Mcam*, *Apln*, *Plaur*). Conversely, C9 *Pdcd10*-ko cells showed up-regulation of a large set of tumor tip cell markers (*Zhao et al., 2018*), which were less modified in C1 and C6 (*Figure 5C*). According to the enhanced expression of the tip cell marker *Plaur* in the *Pdcd10*-ko cells of C9 (as well as in various other clusters), uPAR (encoded by *Plaur*) was highly expressed both in overt lesions and in pseudo-normal vessels of the brains from the *Pdcd10*-ko mice, as shown in *Figure 5D*, and quantified in *Figure 5E*.

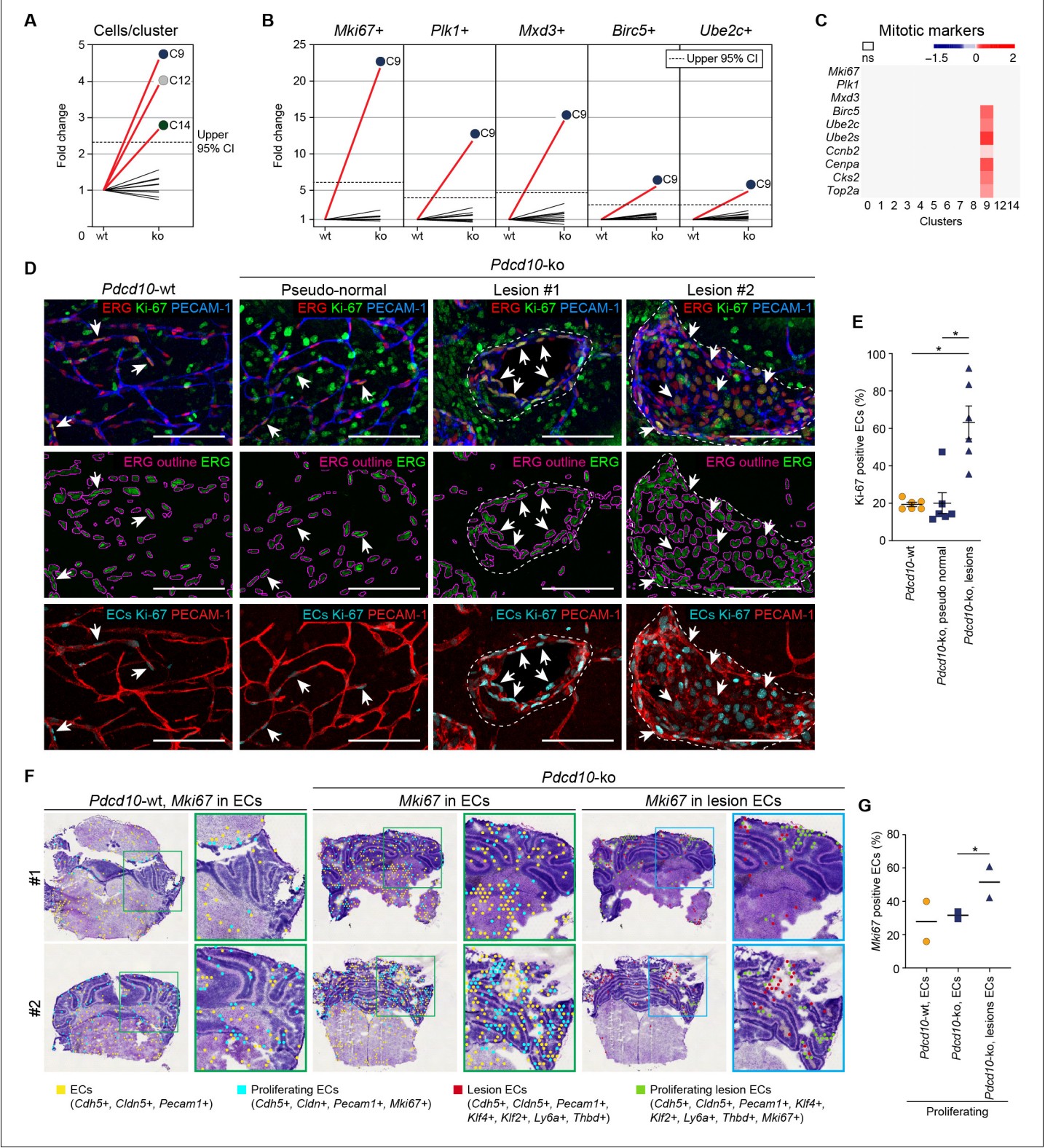

**Figure 4.** *Pdcd10*-ko cells of C9 express the same top mitotic phenotype as overt cavernomas. (A) Plot of fold-change in numbers of *Pdcd10*-ko versus *Pdcd10*-wt cells in each cluster. (B) Plots of fold-change in the percentages (*Pdcd10*-ko vs. *Pdcd10*-wt) of cells positive for *Mki67*, *Plk1*, *Mxd3*, *Birc5*, and *Ube2c* in each cluster (as indicated). Dashed lines in (A) and (B) show upper limit of 95% confidence interval (CI), calculated from the mean fold-changes among the clusters. Red lines, clusters with fold-change >95% CI upper limit. (C) Heatmap of average logFC (padj <0.05) in the *Pdcd10*-ko vs. *Pdcd10*-wt for selected mitotic markers (as indicated). (D) Representative confocal microscopy of *Pdcd10*-wt and *Pdcd10*-ko mouse cerebellum at P8. Upper

*Figure 4 continued on next page*

*Figure 4 continued*

panels: Immunostained for PECAM-1 (pan-endothelial, blue), ERG (pan-endothelial nuclei, red) and Ki-67 (encoded by *Mki67* transcript; mitotic nuclei, green). Central panels: Immunostained for ERG (green) with outlines of the segmented nuclei (magenta). Lower panels: Results of the filtering procedure showing EC-specific immunostaining of Ki-67 (ECs Ki-67, light blue) and PECAM-1 (red) (see Materials and methods). Arrows, mitotic ECs. Scale bars: 100 μm. (E) Quantification of mitotic ECs in the cerebellum, as normal vessels in *Pdcd10-wt*, *Pdcd10-ko* pseudo-normal vessels, and *Pdcd10-ko* lesions (mean ± SEM; *p<0.01; ANOVA followed by Tukey's multiple comparisons). *Pdcd10-wt*, n = 6; *Pdcd10-ko*, n = 6. (F) Visium analysis showing spots positive for ECs (yellow) and lesion EC markers (red), identified as indicated, with co-expression of *Mki67*. Proliferating ECs (light blue) and proliferating lesion ECs (green) containing spots are shown. Higher magnifications of boxed areas are also shown. (G) Quantification of the percentages (%) of proliferating ECs (light blue) and proliferating lesion ECs (green) containing spots as in (F) (Mean; *p<0.01 Fisher's exact test).

The online version of this article includes the following source data and figure supplement(s) for figure 4:

**Source data 1.** Source data file for *Figure 4E*.

**Figure supplement 1.** Validation of Visium analysis and identification of spots co-expressing endothelial and CCM lesion markers.

Furthermore, the Visium analysis allowed identification and quantification of the numbers of spots that co-expressed tip cell markers and either pan-EC markers or lesion-EC markers. Here, both combinations showed strong increases in the *Pdcd10-ko* sections (*Figure 5—figure supplement 1A,B*).

We then localized the tip cells in the *Pdcd10-wt* and *Pdcd10-ko* brain vessels as ECs with filopodia using CD93 membrane staining (*Lugano et al., 2018*). In the *Pdcd10-wt* cerebellum, the tip cells were seen to be located at the ends of and along small vessels, thus marking angiogenic sprouts, as expected in the brains of P8 pups. In the corresponding areas of the *Pdcd10-ko* cerebellum, tip cells were seen at the end of and along pseudo-normal vessels, and also in cavernomas (*Figure 5F*, upper panels). However, considering both pseudo-normal vessels and cavernomas, the densities of the tip cells were similar in the *Pdcd10-wt* and *Pdcd10-ko* brains (*Figure 5G*).

To circumvent the anatomical complexities of the brain vasculature, and to more precisely evaluate the number of tip cells in the *Pdcd10-ko* central nervous system tissue, the retinas were examined. The P8 retina still presents a relatively flat vascular plexus, which facilitates the identification and quantification of the tip cells (*Figure 5F*, bottom panels). The tip cells at the vascular front were significantly denser in the retinas from the *Pdcd10-ko* than for the *Pdcd10-wt* (*Figure 5H*, tip cells at front), while the numbers of filopodia per tip cell were comparable for the *Pdcd10-wt* and *Pdcd10-ko* (*Figure 5H*, filopodia/cell). However, the *Pdcd10-wt* tip cells showed typical elongated cell bodies, while about 15 to 30% of the *Pdcd10-ko* tip cells were not extended and appeared blocked within the dense cell layer (*Figure 5H*, tip-like cells). Similarly, the blocked tip-like cells only represented 10% in the *Pdcd10-wt* (*Figure 5H*, tip-like cells).

Consistent with the impaired tip cell phenotype, radial expansion of the vascular network was significantly inhibited in the *Pdcd10-ko* retina (*Figure 5I*, quantification in *Figure 5J*). It remains to be explored whether these tip-like cells have features of the recently described breach cells of lung tumor vessels (*Goveia et al., 2020*).

Of particular interest, in the *Pdcd10-ko* retina, there were cells with filopodia also in the walls of the veins proximal to the optic nerves (*Figure 5F*, rear, asterisks; *Video 2*). In this region, the tip cells were never observed in the wall of *Pdcd10-wt* veins.

All in all, these data indicate that the *Pdcd10-ko* cells of C9 (i.e. venous/venous capillary) are more mitotic than the respective *Pdcd10-wt* cells. Tip cell traits were also expressed in the

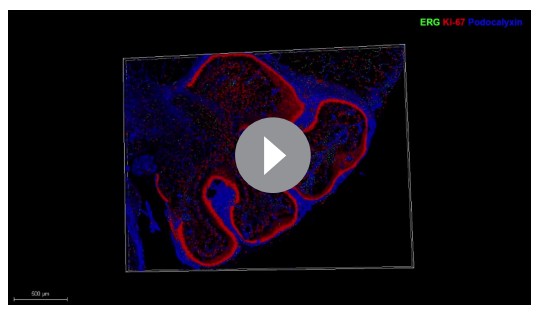

**Video 1.** Three-dimensional rendering of a whole section from a *Pdcd10-ko* cerebellum (100 μm thick) after confocal microscopy. Staining for ERG (green), Ki67 (red), and Podocalyxin (blue) is shown. At 4 s and 20 s, the movie focuses on two examples of mulberry lesions. Ki67 staining in the ERG-positive nuclei is clearly visible (white arrows); to enhance the visibility of Ki67 staining, at each stop the ERG staining was hidden for a few seconds. Two Ki67 positive nuclei in pseudo-normal vessels are visible close to the second lesion shown (yellow arrows, top left and bottom right). Scale bars: as indicated on the bars, as the size changes during the movie.
https://elifesciences.org/articles/61413#video1

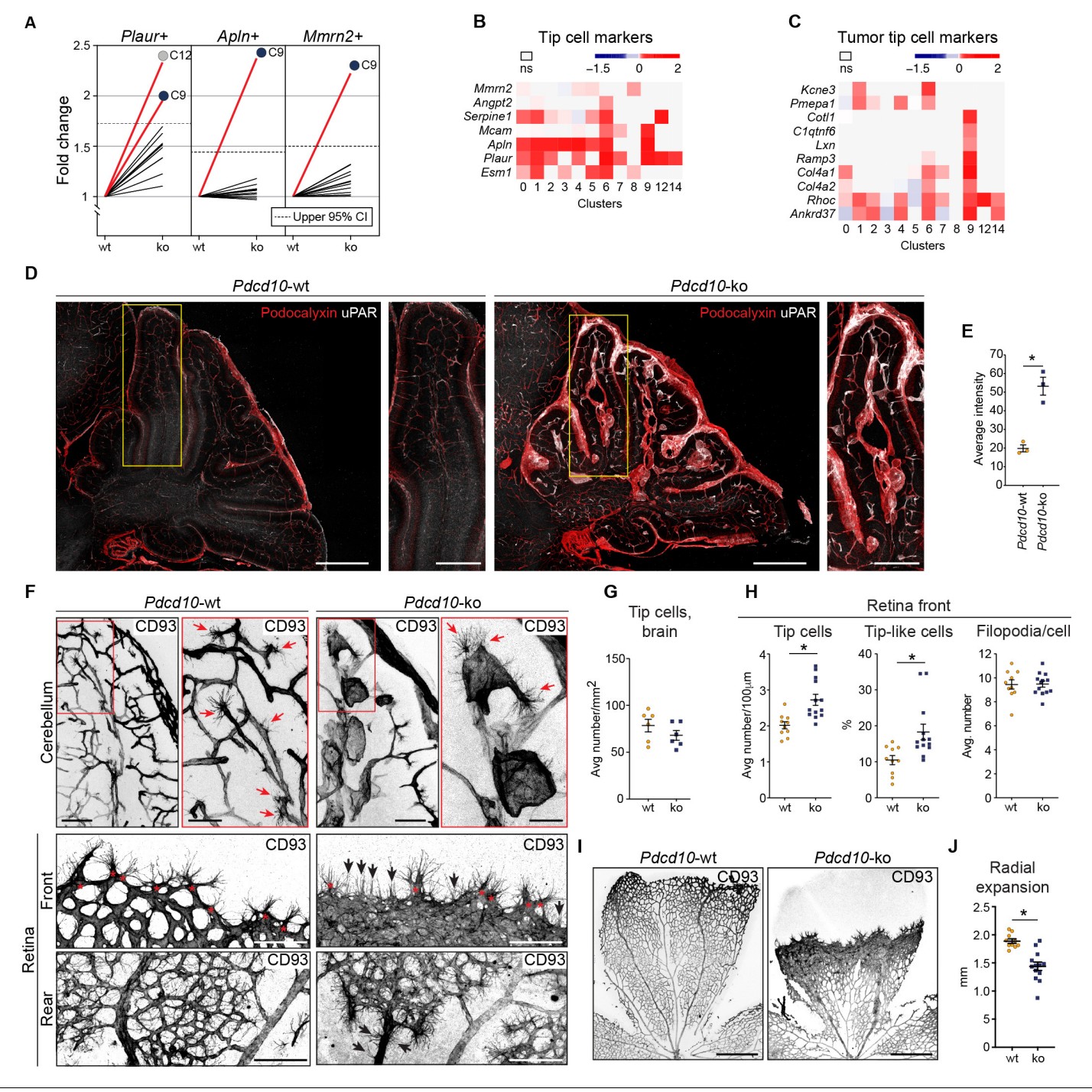

**Figure 5.** *Pdcd10*-ko cells widely enhance the tip cell phenotype with greatest expression for cluster C9. However, such tip cell phenotype is functionally defective. (**A**) Plots of the fold-changes in the proportions (*Pdcd10-ko* vs. *Pdcd10-wt*) of cells positive for *Plaur*, *Apln*, and *Mmrn2* in each cluster. Dashed line shows upper limit of 95% confidence interval (CI), calculated on the mean fold-change among the clusters. Red lines highlight clusters with fold-changes > 95% CI upper limit. (**B**) Heatmap of average logFC (padj <0.05) for *Pdcd10-ko* vs. *Pdcd10-wt* for selected tip cell markers (as indicated). (**C**) Heatmap of average logFC (padj <0.05) for *Pdcd10-ko* vs. *Pdcd10-wt* for selected tumor tip cell markers (as indicated). (**D**) Representative confocal microscopy of *Pdcd10-wt* and *Pdcd10-ko* mouse cerebellum at P8, immunostained for Podocalyxin (pan-endothelial, red) and uPAR (tip cell marker, encoded by *Plaur*, white). Scale bars: main, 500 μm; magnifications, 250 μm. (**E**) Quantification of uPAR staining as in (**D**) (mean ± SEM; *p<0.01; unpaired t-test). *Pdcd10-wt*, n = 3; *Pdcd10-ko*, n = 3. (**F**) Representative confocal microscopy of *Pdcd10-wt* and *Pdcd10-ko* mouse cerebellum (top) and retina (bottom) at P8, immunostained for CD93 (pan-ECs membrane and filopodia, at P8). Retina vessels are shown for the migrating front (Front) and proximal to the optic nerve (Rear). Red arrows and asterisks (in magnifications), tip cells (filopodia-rich cells); black arrows,

*Figure 5 continued on next page*

*Figure 5 continued*

tip-like cells at the migrating front in the *Pdcd10-ko* retina. Tip-like cells were present in the vein proximal to the optic nerve (Rear, black arrows) exclusively for the *Pdcd10-ko*. Scale bars: main, 100 μm. magnifications, 50 μm. (G) Quantification of tip cells in the cerebellum (mean ± SEM; ns; unpaired t-test). *Pdcd10-wt*, n = 6; *Pdcd10-ko*, n = 6. (H) Quantification of tip cells and tip-like cells at the retina front, for tip cell density (mean number/ 100 μm; left), proportion of tip-like cells (%; middle), and number of filopodia/cell (mean ± SEM; right; *p<0.01; unpaired t-test with Welch's correction or Mann-Whitney for tip-like cells). *Pdcd10-wt*, n = 10; *Pdcd10-ko*, n = 13. (I) Representative lower magnification images of whole-mount retina preparations immunostained for CD93. Scale bars: 500 μm. (J) Quantification of radial expansion (mm; mean ± SEM; *p<0.01; unpaired t-test). *Pdcd10-wt*, n = 10; *Pdcd10-ko*, n = 13.

The online version of this article includes the following source data and figure supplement(s) for figure 5:

**Source data 1.** Source data file for *Figure 5E, G, H, and J*.

**Figure supplement 1.** Lesion endothelial cells co-express mitotic and tip cell markers.

*Pdcd10*-ko cells of C9, although they did not show a normal profile. These data were also validated using the Visium analysis, where 15 to 30% of the lesion ECs contained spots that were positive for both tip cell and mitotic-cell markers (*Figure 5—figure supplement 1B,C*).

These data thus support the model that the *Pdcd10*-ko cells in C9 develop aberrant tip cell and mitotic characters, while the *Pdcd10*-ko cells in the other clusters mainly acquire abnormal tip cell traits. This angiogenic (tip cell/mitotic) phenotype is, indeed, defective, as network elongation is impaired, and cell proliferation is enhanced. The combination of high cell proliferation and low cell motility might explain the formation of the cavernomas (see also *Castro et al., 2019*).

## Venous/venous capillary Pdcd10-ko cells represent a reservoir of mitotic and abnormal tip cells that can support defective angiogenesis after Pdcd10 deletion

To further characterize the cellular heterogeneity and the state transitions during lesion formation, endothelial lineage trajectories were de-novo reconstructed based on the clusters that were mainly affected by *Pdcd10* deletion (i.e. C1, C6, C7, C9). Using both *Pdcd10-ko* and *Pdcd10-wt* data, Single-cell Trajectories Reconstruction, Exploration and Mapping (STREAM) (*Chen et al., 2019*) was used to infer a trajectory structure that recapitulated the endothelial lineage differentiation during vascular development, with tip cells (C1, C6) defined on branches S4–S3 and S5–S3, respectively (*Figure 6A*), and venous/venous capillary (C9) and mitotic/capillary (C7) cells on branches S0–S2 and S0–S1 (*Figure 6A*). A fifth branch acted as a connecting segment (*Figure 6A*, S3–S0). Finally, although separated, the S4 and S1 ends appeared to be connected by a small number of C1 cells in the three-dimensional views of the trajectories (*Figure 6—figure supplement 1A,B*). This might explain the apparent separation of C1 in the flat view, and it suggests a continuum between these two branches that is interrupted by the constraints imposed by the two-dimensional view.

In particular, when the *Pdcd10-wt* and *Pdcd10-ko* were separated, we observed the following:

i.  The branch S0–S2 was exclusively composed of *Pdcd10*-ko cells (*Figure 6B*, blue), which mainly belonged to C9 (*Figure 6A*); these thus represent a uniquely formed and potentially pathological endothelial lineage (*Figure 6B*, red arrow, *Figure 6C*).

ii. The connecting branch (S3–S0) contained mainly *Pdcd10*-ko cells, which again mainly belonged to C9, with a negligible contribution of only four *Pdcd10*-wt ECs (*Figure 6B*, black arrow, *Figure 6C*).

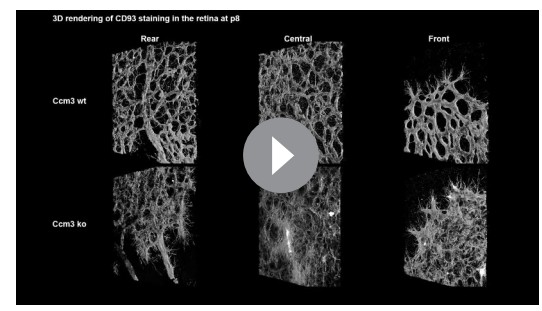

**Video 2.** Three-dimensional rendering of three retinal regions after confocal microscopy. *Pdcd10-wt* (top) and *Pdcd10-ko* (bottom) retinas were stained for the membrane marker CD93 (white). Rear and front regions were acquired in the same areas shown in *Figure 5F*. The central area was from the same retina and shows high density vascular plexus for *Pdcd10-ko*. Increased numbers of tip cells were observed in all of the vascular regions of *Pdcd10-ko* retina.
https://elifesciences.org/articles/61413#video2

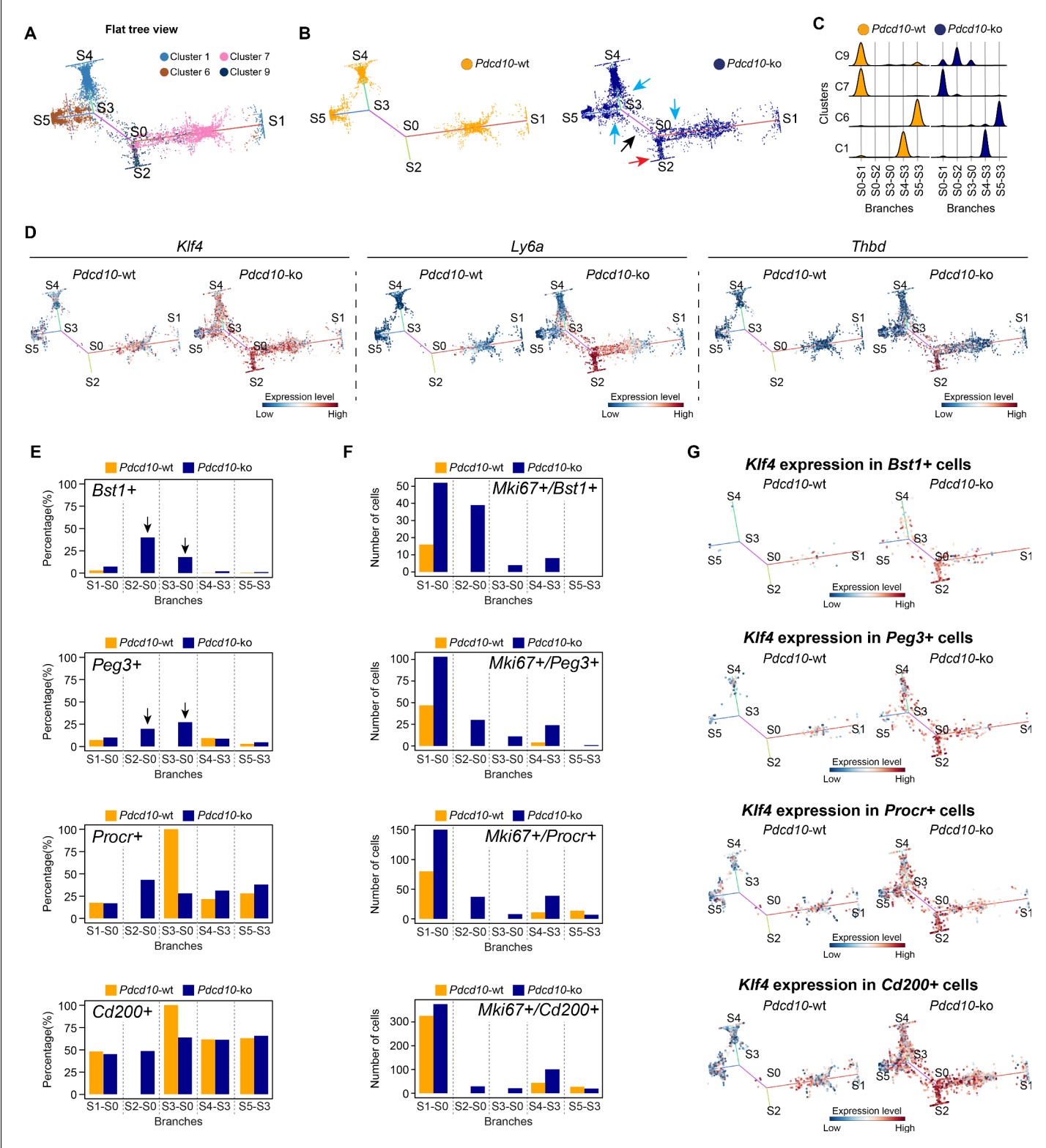

**Figure 6.** STREAM trajectory analysis shows *Pdcd10-ko* venous cluster 9 concentrated in a distinctive branch with mitotic and abnormal tip cell functions. (**A**) Flat tree view of the STREAM trajectory inferred for *Pdcd10*-wt and *Pdcd10*-ko cells of the most affected clusters (C1, C6, C7, C8), color coded as indicated. Branching and end points are numbered sequentially (S0, S1, S2, S3, S4, S5) and each branch is named after the points connected (S0–S1, S0–S2, S4–S3, S3–S0, S5–S3). For corresponding three-dimensional views of the STREAM trajectories see *Figure 6—figure supplement 1A and B*. (**B**) Flat tree view of the trajectory as in (**A**) showing separately *Pdcd10*-wt (orange) and *Pdcd10*-ko (blue) cells. Red and black arrows, *Pdcd10-ko*

*Figure 6 continued on next page*

Figure 6 continued

specific branches (S0–S2, S3–S0, respectively). Light-blue arrows, *Pdcd10*-ko cells showing altered distributions within the other branches. (C) Distributions of *Pdcd10*-wt (orange) and *Pdcd10*- ko (blue) cells according to cluster, for the five branches of the trajectory. *Pdcd10*-ko cells of Cluster 9 (top row, blue peaks) show the most heterogeneous distributions with the cells in branches S0–S1, S0–S2, and S3–S0. (D) Flat tree view showing expression levels in *Pdcd10*-wt (left) and *Pdcd10*-ko (right) cells for three representative lesion markers: *Klf4*, *Ly6a*, and *Thbd*. (E) Plots showing the percentage of cells positive for the progenitor cell markers *Bst1*, *Peg3*, *Procr*, and *Cd200* (as indicated) for *Pdcd10*-wt (orange) and *Pdcd10*-ko (blue) in each branch of the trajectories in (A) and (B). See Materials and Methods for definition of positive cells. (F) Plots showing numbers of cells double positive for *Mki67* and for indicated progenitor cell markers for *Pdcd10*-wt (orange) and *Pdcd10*-ko (blue) in each branch of the trajectories in (A) and (B). See Materials and methods for definition of positive cells. (G) Flat tree view showing expression levels of *Klf4* for *Pdcd10*-wt (left) and *Pdcd10*-ko (right), for each of the progenitor cell subpopulations investigated. In (D and G) dots are color coded for expression levels, as blue, low; red, high. The online version of this article includes the following figure supplement(s) for figure 6:

**Figure supplement 1.** STREAM trajectory analysis of C9 Pdcd10-ko cells from different branches for the expression of functions (GO) and tip, mitotic and transcription factor genes.

**Figure supplement 2.** STREAM trajectory analysis of resident endothelial progenitor cells from different branches for the expression of the CCM-related genes Id1 and Klf2.

iii.  The widespread distribution of the *Pdcd10*-ko cells (*Figure 6B*, blue) compared to the *Pdcd10*-wt cells (*Figure 6B*, orange) on each branch suggested that many *Pdcd10*-ko cells had not reached the end of their lineage differentiation; i.e. the *Pdcd10*-ko cells were either delayed in lineage differentiation or they were on-hold at different immature stages.

iv.  The expression of typical lesion markers *Klf4*, *Ly6a*, and *Thbd* (*Maddaluno et al., 2013*; *Lopez-Ramirez et al., 2019*) was higher for the *Pdcd10*-ko cells specific to the S0–S2 and S3–S0 branches, even if not exclusively limited to these branches (*Figure 6D*).

v.  *Pdcd10*-ko cells that belonged to C9 showed the highest heterogeneity, as these were distributed along three different branches (i.e. S0–S2, S3–S0, S0–S1) (*Figure 6C*). To investigate the high heterogeneity of these *Pdcd10*-ko cells, multiple comparisons were performed, followed by Gene Ontology (GO) enrichment analysis of the DEGs (*Figure 6—figure supplement 1C*). Consistent with the data reported int *Figures 2–5*, the cells in S0–S2 showed strong down-regulation of BBB transporters and increased negative regulation of cell-cell adhesion, compared to those of S0–S1 (*Figure 6—figure supplement 1C*). Moreover, the *Pdcd10*-ko cells in S3–S0 were more involved in sprouting angiogenesis with enhanced membrane and actin dynamics, increased metabolic activity, and dysregulated responses to growth factor stimulation, compared to those of S0–S2 and S0–S1 (*Figure 6—figure supplement 1C*).

vi.  Finally, the *Pdcd10*-ko cells of C9 showed enhanced mitotic and aberrant tip cell characteristics compared to the *Pdcd10*-wt cells in the same cluster (*Figures 4* and *5*). This angiogenic character of mitotic and tip cells was also evident by comparisons of the gene expression levels of a selection of mitotic and tip cell markers for the *Pdcd10*-wt and *Pdcd10*-ko cells in the trajectory (*Figure 6—figure supplement 1D*).

In summary here, the C9 *Pdcd10*-ko cells in the newly formed *Pdcd10*-ko specific branch of S0–S2 highly expressed typical cavernoma markers, they were mainly mitotic, and they also had pathological tip cell traits. All in all, these can be envisaged as ECs of typical cavernoma lesions.

A small proportion of the C9 *Pdcd10*-ko cells in S3–S0 was also apparently more tip cell specialized. However, these cells failed to reach the states of the trajectories where fully specialized tip cells are located (i.e. S4–S3, S5–S3).

## Venous-resident Pdcd10-ko endothelial progenitor cells support the formation of cavernomas

We then investigated the transcription factors that were differentially expressed by the C9 *Pdcd10*-ko cells for the different branches from the STREAM analysis. The *Pdcd10*-ko cells for S0–S2 *versus* S0–S1 showed increased expression of the transcriptional stimulator of progenitor cell proliferation and differentiation, *Tcf15* (*Davies et al., 2013*; *Figure 6—figure supplement 1E*). As previously reported (*Maddaluno et al., 2013*), *Id1* was up-regulated in *Pdcd10*-ko ECs of lesions, and it was consistently strongly increased in the S0–S2 branch (*Figure 6—figure supplement 2A*). Remarkably, *Id1* inhibits *Tcf15*-mediated progenitor cell differentiation, but not *Tcf15*-mediated proliferation of progenitor cells (*Davies et al., 2013*).

Considering that our group recently reported that cavernomas have clonal origins from expansion of *Pdcd10*-ko progenitor cells (*Malinverno et al., 2019*), we investigated the distribution of cells that expressed progenitor cell markers in the STREAM-generated trajectories. We therefore quantified each branch for the proportions of cells positive for *Bst1*, *Peg3*, *Procr*, and *Cd200* (i.e. reported markers of resident endothelial progenitor cells *Malinverno et al., 2017*; *Wakabayashi et al., 2018*; *Yu et al., 2016*). The proportions of *Pdcd10*-ko cells that were positive for *Bst1* or *Peg3* were highly, and specifically, increased in the S0–S2 and S3–S0 branches (*Figure 6E*, arrows). In contrast, the proportions of *Pdcd10*-wt and *Pdcd10*-ko cells positive for *Procr* and *Cd200* were similar across all of the branches of the trajectory (*Figure 6E*, lower panels). The only exception was the high proportion of *Procr*-positive *Pdcd10*-wt cells in the S3–S0 branch, although this might be biased by the very low number of *Pdcd10*-wt cells in this branch (i.e. only four cells).

We then asked whether the increased number of *Pdcd10*-ko progenitor cells might be sustained by their increased proliferation. As shown in *Figure 6F*, co-expression of progenitor cell markers and *Mki67* in *Pdcd10*-wt cells was limited to the cells in the S0–S1 branch (orange bars), where mitotic C7 cells were located (*Figure 6A*). Conversely, there were increased proportions of *Mki67*-positive *Pdcd10*-ko progenitor cells in all of the branches, with the strongest increases seen for S0–S2 (*Figure 6F*, blue bars), compared to the *Pdcd10*-wt. Moreover, the expression levels of *Bst1* were specifically high in the *Pdcd10*-ko cells of the S0–S2 branch, while the expression of the other progenitor cell markers in the *Pdcd10*-ko cells was high, but more similar across all of the branches of the trajectory (*Figure 6—figure supplement 2B*). Interestingly, *Id1* expression was high in *Bst1*-positive cells in the S0–S2 branch, which suggested that differentiation in these resident progenitor cells can be inhibited (*Figure 6—figure supplement 2C*) as mentioned above and shown by *Davies et al., 2013*.

These data indicate that *Bst1*-positive and *Peg3*-positive resident endothelial progenitor cells were specifically concentrated in the branches that were most affected by *Pdcd10* deletion. Interestingly, the key driver genes of the CCM phenotype, *Klf4* and *Klf2*, were highly expressed in *Pdcd10*-ko progenitor cells of the S0–S2 branch that were also positive for *Bst1*, *Peg3*, *Procr*, and *Cd200*, and also for these cells of the other branches (*Figure 6G*; *Figure 6—figure supplement 2D*, respectively). This supports the concept that *Klf4* and *Klf2* are early response genes to *Pdcd10* deletion in these types of resident endothelial progenitor cells.

Colocalization of *Bst1* and the cavernoma transcripts *Klf2*, *Klf4*, and *Ly6a* using Visium (*Figure 7A, B*) showed concentration of *Bst1* progenitor cell transcripts at the level of cavernomas and mulberry lesions (*Figure 7A*). Similarly, we also confirmed the presence of *Bst1*-positive cells in the lesions through in-situ hybridization in brain sections of the *Pdcd10*-wt and *Pdcd10*-ko (*Figure 7C*; *Malinverno et al., 2017*).

In the analysis of the clusters identified in this scRNA-seq analysis, after *Pdcd10* deletion the proportions of *Bst1*-positive cells increased in all of the capillary-vein clusters (C2, C4, C7), with the greatest expression in the venous/venous capillary C9 (*Figure 7D*; *Bst1*-positive cells, 39.9%), which is in agreement with the data reported in *Figure 6E*. In contrast, the proportions of *Bst1*-positive cells did not increase in only the arterial C8 and arterial capillary C3 and C5 *Pdcd10*-ko cells, in comparison with the respective *Pdcd10*-wt (*Figure 7D*). Therefore, *Bst1* appears to be a preferential physiological marker of vein-resident progenitor cells that is particularly increased in venous clusters after *Pdcd10* deletion, with no increase in arterial clusters (*Figure 7D*).

Although *Peg3* appears to be a more general marker of resident endothelial progenitor cells in *Pdcd10*-wt cells, it showed highest expression in venous C9 *Pdcd10*-ko cells (*Peg3*-positive cells vs. wt, 24.0% vs. 9.3%). The venous-capillary C7 *Pdcd10*-ko cells showed the second highest increase (*Figure 7D*). As for *Bst1*, the proportions of *Peg3*-positive cells did not increase in any of the artery and arterial capillary clusters, which suggests that *Pdcd10* deletion can induce endothelial proliferation at different maturation stages. These data are in agreement with the resistance to expression of the *Pdcd10* mutant phenotype of arterial ECs and with the high susceptibility of the venous ECs described above.

The *Procr*-positive and *Cd200*-positive cells (*Figure 7D*) were ubiquitously expressed in the *Pdcd10*-wt venous and arterial clusters, and did not significantly increase after *Pdcd10* deletion at the developmental stage examined (8dpn), in agreement with the trajectory data of *Figure 6E*.

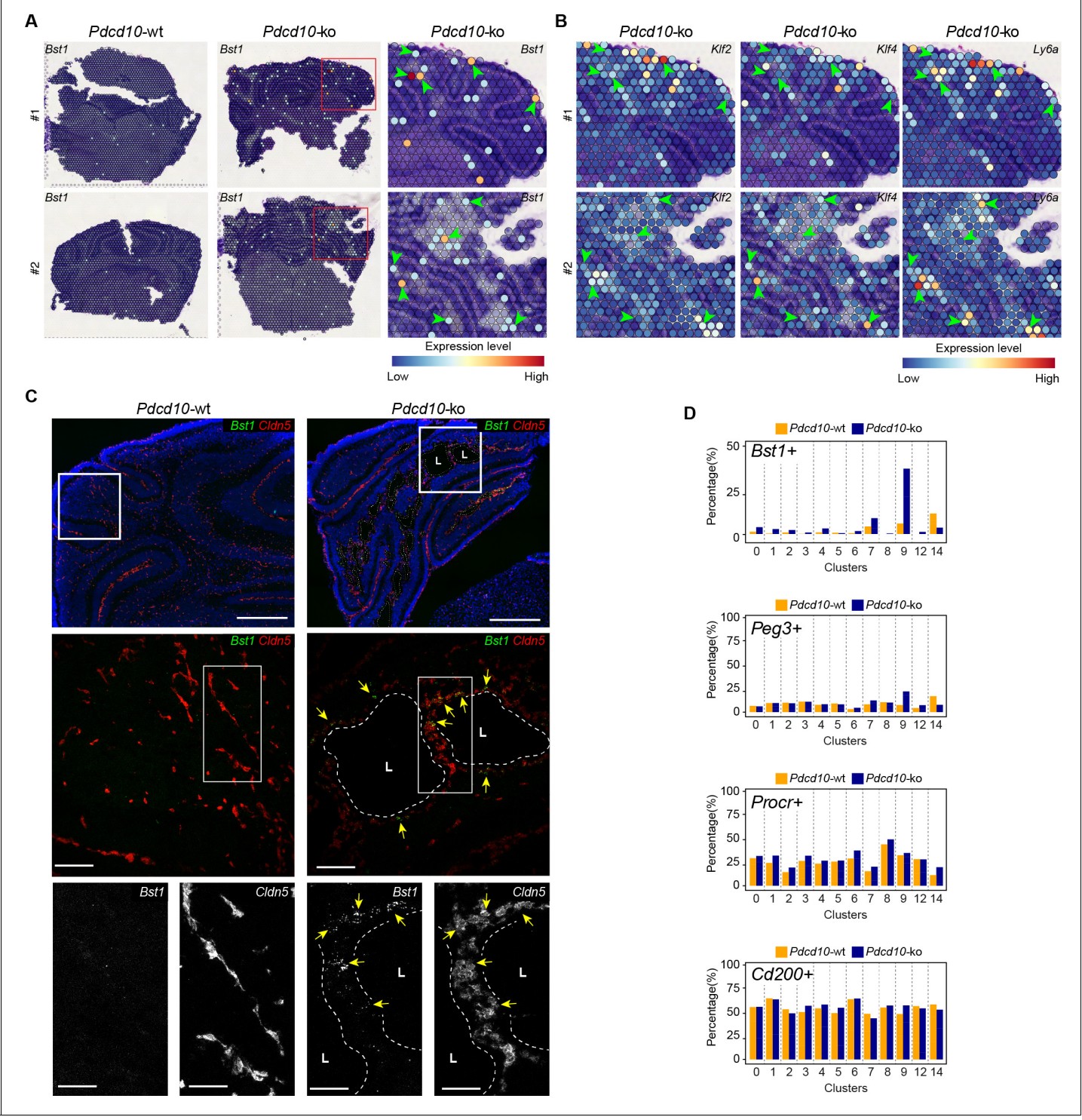

**Figure 7.** Venous-resident *Pdcd10*-ko endothelial progenitor cells support the formation of cavernomas. (A) Expression levels of *Bst1* as measured according to Visium Spatial Transcriptomics. All of the sequenced dots are shown, color coded for expression, as blue, low; red, high. Negative spot outlines are shown without any fill color to allow the visualization of underlying H and E staining (dark and light violet). Both complete sections and magnified boxed areas are shown. Arrowheads, examples of positive spots. (B) Expression levels of representative lesion markers (as indicated) from Visium analysis. For each gene and for each sample, the same magnified areas are shown as in (A). Arrowheads, same spots as indicted in (A). (C) Representative in-situ hybridization for *Bst1* (green) and *Cldn5* (red) using the RNA-Scope assay for *Pdcd10-wt* and *Pdcd10-ko* mouse cerebellum at P8. Three magnifications are shown (increasing top to bottom). Top: DAPI staining (blue) is also shown. Bottom: *Cldn5* and *Bst1* shown separately (white). Yellow arrows points to colocalization of *Cldn5* and *Bst1* signals. Lesion lumens (L) are outlined by white dashed lines. Scale bars: 500 μm (left), 100 μm

*Figure 7 continued on next page*

**Figure 7 continued**

(middle), 50 µm (right). **(D)** Plots of percentages (%) of cells positive for progenitor cell markers *Bst1*, *Peg3*, *Procr*, and *Cd200* (as indicated) for *Pdcd10*-wt (orange) and *Pdcd10*-ko (blue) cells of each cluster. See Materials and Methods for definition of positive cells.

## Discussion

Through single-cell transcriptomic analysis, we have identified here some unique characteristics of ECs that form either normal vessels or cavernomas. Here, we have shown that brain ECs represent a heterogeneous population that is distributed across 13 different clusters on the basis of their gene expression. These cells express highly specialized functions that help to form the vasculature that is most appropriate for the needs of brain vascularization and for the development of the BBB (*Figures 1I* and *8A*). Remarkably, resident progenitor cells were detected in all of the *Pdcd10*-wt clusters, although in different proportions. This indicates that resident progenitor cells are heterogeneous and share several transcriptional features with the ECs of the vessels where they reside (*Figure 8A*). It can be speculated that this 'mimicry' behavior of endothelial-resident progenitors might be determined by the specificity of the different vascular microenvironments, and even that the 'progenitor' status might represent a transient and reversible response to microenvironmental signals, rather than a fate-committed, stable identity. In this study, we have further extended the concept of diversity through consideration of a pathological inherited condition (i.e. CCM) that is characterized by the development of cavernomas in the brain microcirculation. Although there is evidence that in CCM, deficient ECs of venous origin appear to be responsible for the formation of the lesions in the retina, an undisputed demonstration of their origin and of their functional characteristics was still missing (*Abdelilah-Seyfried et al., 2020*; *Boulday et al., 2011*).

We report here that venous ECs are particularly sensitive to *Pdcd10* deletion. Under *Pdcd10*-ko, these venous ECs undergo defective differentiation and angiogenic programs – with abnormal tip cell traits and increased mitosis – that gives rise to cavernous branching and mulberry lesions (*Figure 8B*). This confirms the venous nature of cavernomas at the molecular level, which has been described up to now anatomically. The venous compartment also responds to *Pdcd10* deletion through increased numbers and expression of venous-resident *Bst1*-positive and *Peg3*-positive progenitor cells. These observations confirm our previous study (*Malinverno et al., 2019*), where we showed that the trigger in cavernoma formation is the recruitment of progenitor cells that attract mature ECs into the areas of the lesions. Interestingly, Goveia et al. recently described *Bst1*-positive cells as vein-resident endothelial stem cells that still have unexplored roles in lung tumor vascularization (*Goveia et al., 2020*).

The role of specific progenitors can be different in different pathological contexts. For example, in injury repairing of the infra-renal aorta, *Bst1* did not label highly proliferating cells with regenerating ability (*McDonald et al., 2018*). On the contrary, during liver vascular damage, liver vascular homeostasis, and ischemic hindlimb muscle, *Bst1*-positive stem cells proliferate to reestablish new functional vessels (*Wakabayashi et al., 2018*). Although a comprehensive definition and comparison of the transcriptional and functional specificity of the progenitors that express *Bst1*, *Peg3*, *Cd200*, and *Procr* in different organs at different developmental stages and in different pathological situations is of great interest, the details here remain very limited and deserve further investigation.

The present study confirms and extends our earlier observations, to show that *Pdcd10*-ko cells appear to form malformations due to increased cell growth and defective sprouting of abnormal tip cells (*Figures 4*, *5* and *8*). Another observation derived from the present study is that arterial ECs are resilient to *Pdcd10* deletion. This striking result shows that brain arterial ECs maintain their genetic and functional phenotype also under *Pdcd10* deletion. It is possible that the unique resistance of these cells is induced and maintained by up-regulation of a relatively large set of 'putative defensive' genes that could be modulated directly or indirectly by *Pdcd10* (*Figures 3* and *8*). These data also allow us to conclude that the cells responsible for the formation of cavernomas are of venous and tip cell origin, while ECs of arterial origin are not responsive. Thus, the same mutation and inactivation of *Pdcd10* can result in strongly different responses in ECs of venous and arterial origin, and in the resident progenitor cells.

Interestingly, the absolute level and the relative increases in *Klf2* and *Klf4* in the *Pdcd10*-ko do not directly correlate with the level of response to *Pdcd10* deletion. Almost all of the clusters

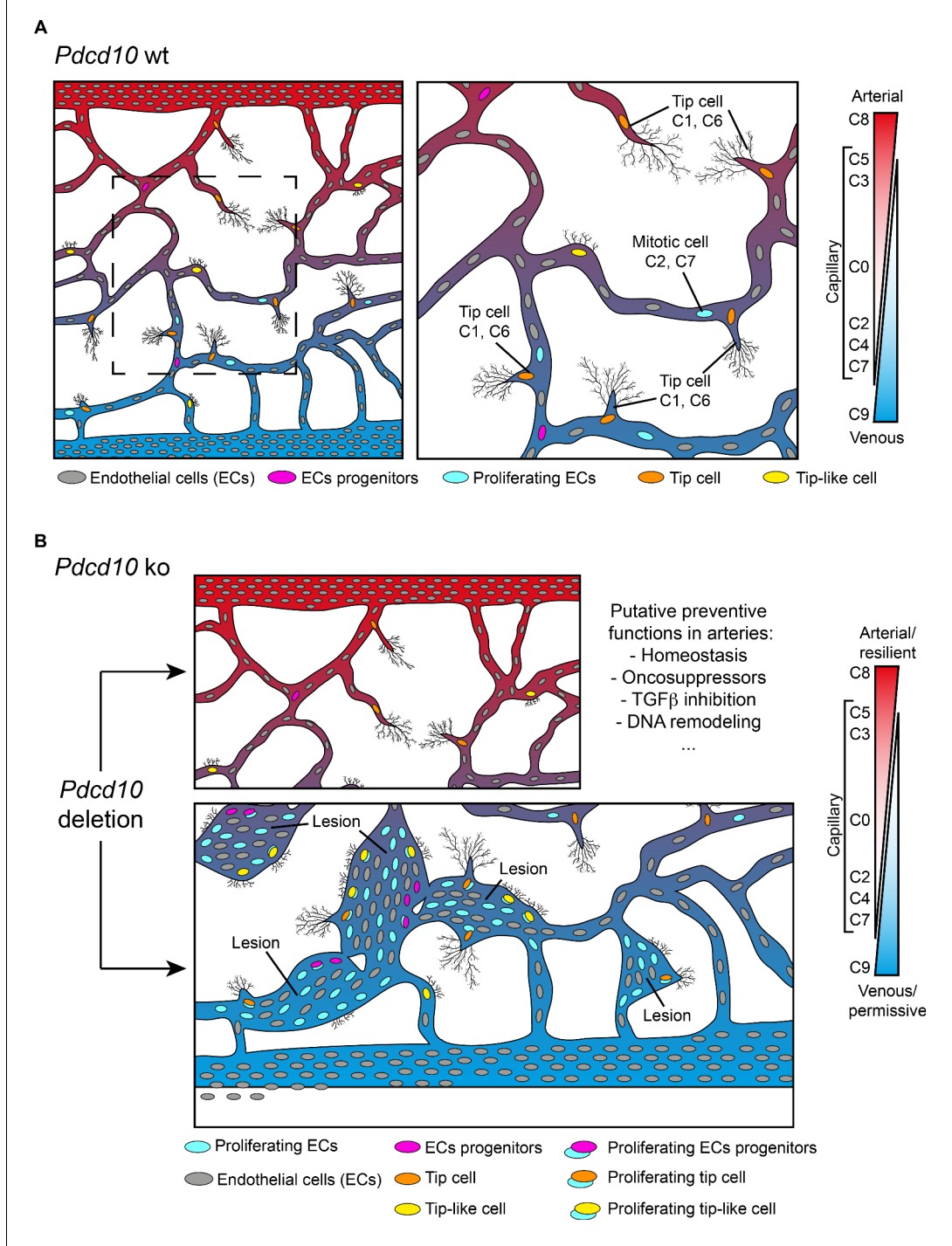

**Figure 8.** Model of brain cavernoma development based on the scRNA-seq analysis of *Pdcd10-ko* brain endothelial cells complemented with the findings obtained by immunofluorescence microscopy and spatial transcriptomics (Visium). Brain ECs (gray) are a heterogeneous population distributed in different clusters on the basis of their gene expression in both *Pdcd10-wt* (A) and *Pdcd10-ko* (B). Besides arterial, arterial capillary, venous, venous-capillary also tip cell and mitotic endothelial clusters could be detected at the developmental stage examined, P8 (orange and light blue , respectively). Besides, tip-like cells could be recognized morphologically (yellow). They are significantly increased in *Pdcd10-ko*, but their transcriptomic features remain to be studied. Resident endothelial cell progenitor cells (pink) were detected in all clusters, although in different proportions, which indicates that they are heterogeneous and share several transcriptional features with the ECs of the vessels where they reside. In (A) dashed square on the left is enlarged for the image on the right. (B) Although *Pdcd10* deletion takes place comparably in arterial and venous ECs, only venous ECs can respond to *Pdcd10* deletion. *Pdcd10-ko* venous cells follow defective differentiation and angiogenic programs, with abnormal tip cell traits and increased mitosis, giving rise to cavernous branching and mulberry lesions. Moreover, the venous compartment responds to *Pdcd10* deletion with increased numbers and

*Figure 8 continued on next page*

*Figure 8 continued*

expression levels of venous resident endothelial cell progenitors. Arterial ECs are instead resilient to *Pdcd10* deletion. It appears that the resistance of these cells is maintained by a relatively large set of putative defensive genes, some of which modulated by *Pdcd10 deletion.* In parallel, endothelial progenitor cells within arterial cluster C8 neither increased in number nor were transcriptionally modified by *Pdcd10* deletion.

identified show up-regulation of both *Klf2* and *Klf4* in *Pdcd10*-ko cells, in comparison to the respective *Pdcd10*-wt. However, while *Pdcd10*- ko cells in all of the clusters are modified, only a few clusters are particularly responsive to the mutation. This indicates that while *Klf2* and *Klf4* are necessary for expression of the mutant phenotype (*Zhou et al., 2016*; *Cuttano et al., 2016*), they are not sufficient. A specific transcriptional landscape appears necessary to allow the full transformation. This is particularly evident in the arterial cluster that express the highest levels of *Klf2* and *Klf4* but does not develop the *Pdcd10* phenotype.

Overall, this study adds several new aspects to the evolution of CCM and identifies the venous EC clusters and their related progenitor cells as the main cell types responsible for cavernoma development. This study thus opens new perspectives in our understanding of the development of cavernous malformations, and it should thus help in the development of novel and better-targeted therapeutics. Here we represents a public source for data exploration of the CCM and the physiological endothelium, as the data can be further explored in the online database https://edgroup.shinyapps.io/MapCcm3EC/.

# Materials and methods

## Key resources table

| Reagent type (species) or resource | Designation | Source or reference | Identifiers | Additional information |
|---|---|---|---|---|
| Strains, strain backgrounds (mice) | *Cdh5*(PAC)-Cre-ER^T2^, C57BL/6 background | *Wang et al., 2010* | | |
| Strains, strain backgrounds (mice) | *Cdh5*(PAC)-Cre-ER^T2^/*Ccm3f/f*, C57BL/6 background | *Bravi et al., 2015* | | |
| Strains, strain backgrounds (mice) | *Cldn5*(BAC)-GFP (Tg(*Cldn5*-GFP)Cbet/U), C57BL/6 background | *Honkura et al., 2018* | | |
| Strains, strain backgrounds (mice) | *Cdh5*(PAC)-Cre-ER^T2^/ *Ccm3f/f/ Cldn5*(BAC)-GFP, C57BL/6 background | This paper | | |
| Antibody | Anti-CD93 (sheep polyclonal) | R and D | AF1696, RRID:AB_354937 | IF (1:500) |
| Antibody | Anti-Erg (rabbit monoclonal) | Abcam | Ab92513, RRID:AB_2630401 | IF (1:400) |
| Antibody | Anti-Ki67 (rat monoclonal) | Invitrogen | 14-5698-82, RRID:AB_10854564 | IF (1:200) |
| Antibody | Anti-Podocalyxin (goat polyclonal) | R and D | AF1556, RRID:AB_354858 | IF (1:400) |
| Antibody | Anti-CoupTFII (rabbit polyclonal) | LSBio | LS-C356225 | IF (1:200) |
| Antibody | Anti-PECAM-1 (rat monoclonal) | BD | 55369, RRID:AB_394815 | IF (1:200) |
| Antibody | Anti-PECAM-1 (armenian hamster monoclonal) | Millipore | MAB1398z, RRID:AB_94207 | IF (1:500) |

*Continued on next page*

*Continued*

| Reagent type (species) or resource | Designation | Source or reference | Identifiers | Additional information |
|---|---|---|---|---|
| Antibody | Anti-Claudin-5 (mouse monoclonal, Alexa fluor 488-conjugated) | Invitrogen | 352588, RRID:AB_2532189 | IF (1:200) |
| Antibody | Anti-VE-cadherin (rat monoclonal) | BD | 550548, RRID:AB_2244723 | IF (1:200) |
| Antibody | Anti-Cingulin (rabbit polyclonal) | Invitrogen | 36–4401 | IF (1:500) |
| Antibody | Alexa Fluor 488, 555 and 647 donkey secondary antibodies | Jackson laboratories | | IF (1:400) |
| Antibody | Alexa Fluor 488 conjugated donkey anti-mouse | Jackson laboratories | | IF (1:400) |
| Antibody | Anti-uPAR antibody | *Tjwa et al., 2009* | | IF (1:200) |
| Commercial assays or kits | 3-Plex positive controls | ACD | #320881 | for detecting Polr2a, PPIB, UBC mRNA |
| Commercial assays or kits | 3-Plex negative controls | ACD | #320871 | for detecting DapB mRNA of *Bacillus subtilis* |
| Commercial assays or kits | Multiplex fluorescent Reagent Kit v2 | ACD | 323100 | |
| Commercial assays or kits | Cldn5 | ACD | 491611-C2 | |
| Commercial assays or kits | Bst1 | ACD | 559841-C3 | |
| Commercial assays or kits | CD45 MicroBeads | Miltenyi Biotech | 130-052-301 | |
| Commercial assays or kits | CD31 MicroBeads | Miltenyi Biotech | 130-097-418, RRID:AB_2814657 | |
| Commercial assays or kits | Adult Brain Dissociation kits | Miltenyi Biotech | 130-107-677 | |
| Commercial assays or kits | Single Cell 3' Reagent kits v2 | 10 × genomics | PN-120237 | |
| Instruments | GentleMACS Octo Dissociator | Miltenyi Biotech | 130-095-937 | |
| Instruments | Chromium controller | 10 × genomics | NA | |
| Instruments | NovaSeq 6000 system | Illumina | RRID:SCR_016387 | |
| Instruments | Vibratome | Leica | VT1200s, RRID:SCR_018453 | |
| Instruments | Sp8 confocal microscope | Leica | | |
| Instruments | Cryostat | Thermo Scientific | CryoStar NX50 | |
| Instruments | High Sensitivity RNA ScreenTape | Agilent | | |
| Others | Visium Spatial Gene Expression slides | 10 × genomics | PN-2000233 | |

*Continued on next page*

*Continued*

| Reagent type (species) or resource | Designation | Source or reference | Identifiers | Additional information |
|---|---|---|---|---|
| Others | Visium Tissue Optimization Slides and reagents | 10 × genomics | PN-1000193 | |
| Software, algorithms | Fiji | open source, http://fiji.sc/ | RRID:SCR_002285 | |
| Software, algorithms | Cell Ranger | 10 × genomics | v2.1.0, RRID:SCR_017344 | |
| Software, algorithms | R package Seurat | *Butler et al., 2018*; *Stuart et al., 2019* | v3.1, RRID:SCR_016341 | |
| Software, algorithms | R package clustree | *Zappia and Oshlack, 2018* | v0.4.2, RRID:SCR_016293 | |
| Software, algorithms | R package EnrichR | *Chen et al., 2013*; *Kuleshov et al., 2016* | v2.1, RRID:SCR_001575 | https://amp.pharm.mssm.edu/Enrichr/ |
| Software, algorithms | GSEA | *Liberzon et al., 2015*; *Subramanian et al., 2005* | v2.2.3, RRID:SCR_003199 | https://www.gsea-msigdb.org/gsea/msigdb/annotate.jsp |
| Software, algorithms | STREAM | *Chen et al., 2019* | v0.4.1 | |
| Software, algorithms | Space Ranger | 10 × genomics | v1.0.0 | |
| Software, algorithms | Loupe cell browser | 10 × genomics | v4.0, RRID:SCR_018555 | |

## Study design

The research objective was to define and characterize the endothelial subpopulation(s) that form lesions in CCM using single-cell RNA sequencing (scRNA-seq) technology. As previously described (*Bravi et al., 2015*; *Malinverno et al., 2019*) we have extensive experience with the P8 neonatal CCM mouse model, upon which the sample sizes were estimated here. ScRNA-seq was performed on littermates, with two mice per conditions. A total of 32,261 cells were sequenced with good quality reads. Visium spatial transcriptomics was performed on littermates, with two mice per condition, one section per mouse. uPAR staining and *Bst1* RNA-scope were performed on littermates, with three mice per condition. All of the other experiments were performed twice with different litters and for n ≥ 6 (see details in Figure legends). No data inclusion or exclusion criteria were predefined or applied. No outliers were defined or excluded. Given the early time points before weaning and sexual maturity, no attempt was made to distinguish or segregate the data based on gender. Injection of tamoxifen was performed 1 day after birth without knowledge of the genotypes.

## Statistical analysis

Unless differently specified, statistical analysis was performed as follow using GraphPad Prism 8. The data were tested for normality using Shapiro-Wilk tests. The data showing normal distributions were analyzed using unpaired t-tests or one-way analysis of variance (ANOVA) followed by *post-hoc* analysis (Tukey's or Sidak's multiple comparisons tests), as specified in the relevant Figure legends. Non-parametric data were analyzed using Mann-Whitney tests. For each experiment, the number of samples is indicated in the relevant Figure legends or in the corresponding Material and Methods section. More detailed information can be found in *Supplementary file 9*.

## Murine models

The following mouse strains were used for immunofluorescence analysis: Cdh5(PAC)-Cre-ER[T2]/ *Ccm3*[f/f] mice in which *Ccm3*[f/f] mice with exons 4–5 of the *Pdcd10 (Ccm3)* gene flanked by loxP sites

(Taconic Artemis GmbH) were bred with *Cdh5*(PAC)-Cre-ER^T2 mice to obtain endothelial-specific and tamoxifen-inducible loss of function of the *Pdcd10* (*Ccm3*) gene, as previously described (*Bravi et al., 2015*). The Cdh5(PAC)-Cre-ER^T2 mouse line was kindly provided by R.H. Adams (Department of Tissue Morphogenesis, Faculty of Medicine, Max Planck Institute for Molecular Biomedicine University of Münster, Münster, Germany) (*Wang et al., 2010*). The genotype of the *Pdcd10* (*Ccm3*) ko mice was *Ccm3^f/f*; *Cdh5*(PAC)-Cre-ER^T2+. *Pdcd10* (*Ccm3*) wt mice were *Ccm3^f/f*; *Cdh5*(PAC)-Cre-ER^T2-.

The following mouse strains were used for scRNA-seq and Visium spatial transcriptomics: *Cdh5* (PAC)-Cre-ER^T2/*Ccm3^f/f*/*Cldn5*(BAC)-GFP mice were generated by crossing *Cdh5*(PAC)-Cre-ER^T2/ *Ccm3^f/f* mice with the *Cldn5*(BAC)-GFP (Tg(*Cldn5*-GFP)Cbet/U) reporter mice (kindly provided by C. Betsholtz; Department of Immunology, Genetics and Pathology, Rudbeck Laboratory, Uppsala University, Uppsala, Sweden) to obtain endothelial-specific expression of GFP (*Honkura et al., 2018*). The genotype of the *Pdcd10* (*Ccm3*) ko mice was *Ccm3^f/f*; *Cdh5(PAC)*-Cre-ER^T2+; *Cldn5*(BAC)-GFP +. *Pdcd10* (*Ccm3*) wt mice were *Ccm3^f/f*; *Cdh5(PAC)*-Cre-ER^T2-; *Cldn5*(BAC)-GFP+.

The mice were all bred on and back-crossed into the C57BL/6 background.

## Tamoxifen treatment

Tamoxifen was dissolved in 10% ethanol-sunflower oil (10 mg/mL) and administered to the mice to induce Cre activity and genetic modifications. In all of the experiments, mouse pups received a single intragastric injection of 100 µg tamoxifen at P1(23, 34), and the tissues were harvested at P8. Two mice of the *Pdcd10-wt* and two mice of the *Pdcd10-ko* from the same litter were used for the single cell datasets. No statistical method was used to predetermine sample size.

The experimental animal protocols were approved by the Uppsala Ethical Committee on Animal Research (permit number C145/15) and the Italian Ministry of Health. Animal procedures were performed in accordance with the Institutional Animal Care and Use Committee (IACUC) and in compliance with the guidelines established in the Principles of Laboratory Animal Care (directive 86/609/ EEC).

## Brain endothelial single-cell isolation

The brains of *Pdcd10-ko* and *Pdcd10-wt* mice were collected and the olfactory bulbs were removed. The brain single cell isolation was performed following the manufacturer's instructions for Adult Brain Dissociation kits (Miltenyi Biotech, 130-107-677). In brief, the brain tissues were first mechanically and enzymatically dissociated using gentleMACS Octo Dissociator (Miltenyi Biotech), then myelin, cell debris, and erythrocytes were removed. ECs were enriched by depletion of CD45-positive cells using CD45 MicroBeads (Miltenyi Biotech, #130-052-301) and selection of CD31-positive cells using CD31 MicroBeads (Miltenyi Biotech, #130-097-418). Then, the collected brain endothelial single cells were processed for scRNA-seq.

## Single-cell library preparation, sequencing and data pre-processing

The scRNA-seq libraries were prepared using the 10× Genomics Chromium system (10× Genomics), following the manufacturer instructions for the Single Cell 3' Reagent kits v2 (10× Genomics, #PN-120237). A total of 10,000 cells were targeted for capture from each mouse sample. Subsequently, the libraries were pooled and sequenced on a NovaSeq 6000 system (Illumina) with an aimed depth of 50,000 reads per cell. The raw data were processed using the analysis pipeline Cell Ranger (10× Genomics, 2.1.0 version) for sample demultiplexing, barcode processing, reads alignment to mouse reference genome (mm10) and single-cell 3' gene counting.

## Single-cell RNA-seq data analysis

### Integrated analysis

The R package Seurat (v.3.1) (*Butler et al., 2018*; *Stuart et al., 2019*) was used to further analyze the pre-processed data from the Cell Ranger pipeline. The quality controls were performed after first merging the data by sample genotype. Cells with detected genes outside of 2 standard deviations (±2 SD) of all of the detected genes were filtered out, and cells with >5% mitochondrial counts were filtered out. The gene expression data of *Pdcd10-wt* and *Pdcd10-ko* were first log normalized to a scale factor of 10,000 and then regressed on the total number of molecules detected per cell (nUMI)

using the ScaleData function. Then, variable genes were identified using the FindVariableFeatures function based on the variance stabilizing transformation method. Afterward, the *Pdcd10-wt* and *Pdcd10-ko* data were merged for the integrated analysis. In the integrated analysis, anchors with pairwise correspondence between individual cells of two genotypes were first identified using the FindIntegrationAnchors function. Then the merged data were scaled and used for principal component analysis. For the clustering analysis, 30 principal components (PCs) were used as the input to a graph-based clustering function in Seurat, FindClusters. The resolution parameter for FindClusters, which determined the number of returned clusters, was decided based on how the samples moved as the number of clusters increased, using the R package clustree (*Zappia and Oshlack, 2018*). Finally, UMAP, a non-linear dimensionality reduction method, was used to visualize and explore cells and clusters in the dataset (*Figure 1D* and *Figure 1—figure supplement 3*). To identify canonical marker genes that were conserved across genotypes and define the clusters, FindConservedMarkers function was used. To identify the DEGs between genotypes or selected groups, the FindMarkers function was used.

## Cell-type annotation of clusters

To assign putative cell type identities to clusters, a list of key genes that define different endothelial subpopulations were collected based on previous publications (*Supplementary file 1*). Then the number of key genes that were included in the canonical marker gene list of each cluster was analyzed (*Supplementary file 7*; avg_logFC > 0 and minimump_p_val < 0.05). A cluster was annotated by a cell type if this cluster had ≥50% key genes of this cell type as the canonical cluster marker genes.

## Selection and evaluation of clustering resolution

A set of resolution parameters that ranged from 0.01 to 1 was used to generate different numbers of clusters at the clustering step during the integrated analysis. R package clustree (*Zappia and Oshlack, 2018*) was used to visualize the relationships between clusters at multiple levels of resolution in a clustering tree (*Figure 1—figure supplement 3*). The selection of the resolution was based on the identification of stable and over-clustered regions in the clustering tree. A stable region was defined as a range of resolution where no additional sub-branching took place in between, and where the total number of clusters remained constant. An over-clustered region normally began with low in-proportion edges, with new clusters then forming from multiple parent clusters. Finally, the highest resolution that remained in a stable and not over-clustered region was selected to form the clusters.

To further evaluate how likely the clusters generated from the selected resolution that was actually present in the dataset, we overlaid the prevalent cell type identity information of the clusters onto the cluster tree (*Figure 1—figure supplement 4*). The lower the resolution when two lineages branched, the higher the chance that the derived clusters were different.

## Comparative analysis

The DEGs between *Pdcd10-wt* and *Pdcd10-ko* were defined with a cut-off of adjusted p-values<0.05 (*Supplementary file 8*; p_val_adj < 0.05). The statistical summaries of the DEGs in each cluster were visualized graphically (*Figure 2A*). The regulation of *Klf2* and *Klf4* between *Pdcd10-wt* and *Pdcd10-ko* were indicated by log-fold changes, as illustrated in the plots (*Figure 2B*). The differences in the regulation of CCM lesion genes, BBB-related genes, and mitotic, tip cell and tumor tip cell markers are shown as heatmaps (*Figures 2C,G*, *4C* and *5B,C*), in which the log-fold changes in gene expression levels between *Pdcd10-wt* and *Pdcd10-ko* are shown.

## Over-representation analysis

The EnrichR web server (*Chen et al., 2013*; *Kuleshov et al., 2016*) (https://amp.pharm.mssm.edu/Enrichr/) (a gene set enrichment analysis web server), was used to investigate the enrichment of CCM-related genes in the significantly up-regulated genes (up-DEGs; *Supplementary file 8*; padj <0.05) in *Pdcd10-ko* comparing with *Pdcd10-wt* in each cluster. Enrichment analysis was performed using EnrichR and its Rare Diseases GeneRIF (*Mitchell et al., 2003*) library. This library summarizes the rare diseases related genes into gene sets based on their GeneRIF (Gene Reference Into Function) annotation from NCBI. For each cluster, the number of DEGs that specifically overlapped

the 'cerebral cavernous malformation (CCM)' gene set is summarized in *supplementary file 3* and shown graphically (*Figure 2D*). The 'cerebral cavernous malformation (CCM)' gene set size is reported at the top of the chart.

The tool 'compute overlaps' from web application GSEA (*Liberzon et al., 2015*; *Subramanian et al., 2005*) (https://www.gsea-msigdb.org/gsea/msigdb/annotate.jsp) was used for over-representation analysis. The significantly up-regulated or down-regulated genes (*Supplementary file 8*, average |logFC| ≥ 0.3 and padj <0.05) between *Pdcd10-wt* and *Pdcd10-ko* in each cluster were analyzed for enriched gene sets in the collection 'Hallmark' from MSigDB (v7.0). The significantly enriched gene sets (p<0.05, q > 0.25, odds of overlapped genes to genes in the gene-set ≥4, overlapped genes > 2) are shown graphically (*Figure 2E*).

### Cluster 8 analysis

Unique canonical marker genes. The canonical marker genes of cluster 8 (C8; *Supplementary file 7*) were considered unique if they were identified as canonical marker genes only for C8 and fulfilled the following criteria (Specificity): adjusted p-value (p_val_adj)<0.05 and average logFC (avg_logFC) >0.25.

Unique canonical marker gene annotation. The unique canonical marker genes of C8 were manually annotated using the Mouse Genome Informatics (http://www.informatics.jax.org/) and NIH National Library of Medicine (PubMed, https://pubmed.ncbi.nlm.nih.gov/) databases. The keywords used for the PubMed search were 'gene/protein name/s' (as in Mouse Genome Informatics) and 'endothelial cells'. If the search gave no results only the 'gene/protein name/s' was used. References for the functional annotations summarized in *Figure 3* are reported in *Supplementary file 5* and *6*.

Unique DEGs (*Pdcd10-ko* vs. wt). The significantly differentially expressed genes (DEGs) between *Pdcd10-ko* and wt in each cluster were identified using FindMarkers function from Seurat R package with a cutoff of adjusted p-values<0.05 and average |logFC| > 0.2. If the significant DEGs were only identified in cluster eight and not in any other clusters, they were considered as unique DEGs for cluster 8.

### Cell population distribution by genotype

The cell ratio between *Pdcd10-wt* and *Pdcd10-ko* in each cluster was analyzed and illustrated graphically (*Figure 4A*). Statistical analysis of the confidence interval was performed using cell ratios from all of the clusters. The dashed line in the line plot indicates the upper limit of the 95% confidence interval (CI) in the dataset. A cell ratio above the upper limit means that the proportion of *Pdcd10-ko* cells in this cluster was much larger compared to that of the other clusters.

Mitotic cells. The expression levels (>0) of mitotic markers *Mki67*, *Plk1*, *Mxd3*, *Birc5*, and *Ube2* were used for selection of the different mitotic cell populations. Then for each mitotic cell population, the cell ratios between *Pdcd10-wt* and *Pdcd10-ko* in each cluster were analyzed and used for confidence interval analysis, as described above (*Figure 4B*).

Tip cells. The expression levels (>0) of tip cell markers *Plaur*, *Apln*, and *Mmrm2* were used for selecting the different tip cell populations. Then, in each tip cell population, the cell ratio between *Pdcd10-wt* and *Pdcd10-ko* in each cluster was analyzed and used for confidence interval analysis, as described above (*Figure 5A*).

Progenitor cells. The expression level quartiles of progenitor markers *Bst1*, *Peg3*, *Cd200*, and *Procr* in all the cells were calculated respectively. Then the first quartile (Q1) of each marker was used for selection of the corresponding progenitor cell population (expression >Q1). The proportion of progenitor cell populations in either *Pdcd10-wt* or *Pdcd10-ko* of each cluster is shown graphically (*Figure 7D*).

### Trajectory analysis

STREAM (*Chen et al., 2019*) is an interactive pipeline for reconstructing developmental trajectories from single-cell data, and it was used here to reorder the cells along the lesion developmental lineages. The normalized expression data from cells in clusters 1, 6, 7, and nine that were calculated in Seurat were passed directly into STREAM.

STREAM performed three main steps to infer a trajectory structure: selection of informative genes, dimensionality reduction, and simultaneous tree structure learning and fitting using

ElPiGraph. The structure of the trajectory was plotted in three-dimensional tree views (*Figure 6—figure supplement 1A,B*) and 2D flat tree views (*Figure 6A,B*), where the branches are represented as straight lines and each circle represents a single cell. The lengths of the branches and the distances between the cells and their assigned branches are preserved from the space where trajectories were inferred.

To identify DEGs among different cell populations in the trajectory, the FindMarkers function in Seurat was applied and significant DEGs were selected using the following criteria: adjusted p-value (pValAdj) <0.05 and average log-fold change (|Avg LogFC|)>0.25. The GO enrichment analysis of significant DEGs was performed using EnrichR, the enriched GO terms were screened by satisfying an adjusted p-value<0.05.

The expression level quartiles of the progenitor markers, *Bst1, Peg3, Cd200, and Procr*, in all the cells included in the trajectory analysis were calculated respectively. Then the first quartile (Q1) of each marker was used for selection of the corresponding progenitor cell population (expression >Q1). The proportion of progenitor cell populations in either *Pdcd10-wt* or *Pdcd10-ko* of each trajectory branch is shown graphically (*Figure 6E*).

The mitotic progenitor populations were further selected based on the expression levels (>0) of a mitotic marker Mki67 in the progenitor cell populations (expression >Q1, *Figure 6E*). The number of mitotic progenitor cells in either *Pdcd10-wt* or *Pdcd10-ko* of each trajectory branch were shown graphically (*Figure 6F*).

## Immunofluorescence

All immunostaining of both the control and mutant samples was carried out simultaneously and under the same conditions. Tissues were prepared and processed for immunohistochemical analysis as described previously (*Bravi et al., 2015*; *Corada et al., 2013*). Briefly, mice were anesthetized by intraperitoneal injection of Avertin 20 mg/kg (Sigma, T48402), and perfused with 1% paraformaldehyde (PFA) in phosphate-buffered saline (PBS).

The mouse brains were carefully dissected out from the skull and post-fixed overnight by immersion in 4% PFA at 4℃. The next day, they were washed in PBS and processed for sectioning as follows. Brains were embedded in 3% low-melting-point agarose, sectioned with a vibratome (100 mm thickness; VT1200s; Leica), and immune stained. Sagittal vibratome sections were incubated in PBST solution (PBS with 0.3% Triton X-100) supplemented with 5% donkey serum and containing the primary antibodies, overnight at 4℃. This was followed by over-day washes with PBST and then the secondary antibody solutions, overnight at 4℃. Sections were then washed over-day, post-fixed for 2 min in 1% PFA, and mounted in Vectashield that contained DAPI (H-1200; Vector). Detection of Claudin-5, Zo1, and Cingulin was carried out as above, with the only exception that brains were post-fixed overnight in 100% methanol at 4℃ (instead of 4% PFA) and rehydrated for 1 hr in PBS before sectioning.

The mouse eyes were collected and fixed in 4% PFA in PBS for 2 hr at 4℃. Retinas were dissected as described previously (*Corada et al., 2013*) and incubated at 4℃ overnight in primary antibodies diluted in PBSTC buffer (PBS with 0.5% Triton X-100, 0.1 mM CaCl$_2$) supplemented with 5% donkey serum, followed by the suitable species-specific Alexa Fluor-conjugated secondary antibody staining, overnight at 4℃.

## Antibodies

The following antibodies were used: anti-CD93 (R and D, AF1696, 1:500); anti-Erg (Abcam, Ab92513, 1:400); anti-Ki67 (Invitrogen, 14-5698-82, 1:200); anti-Podocalyxin (R and D, AF1556, 1:400); anti-CoupTFII (LSBio, LS-C356225, 1:200); anti-PECAM-1 (BD, 55370, 1:200 or Millipore, MAB1398z, 1:500); anti-Claudin-5 (Invitrogen, 352588, 1:200); anti-VE-cadherin (BD, 550548, 1:200); and anti-Cingulin (Invitrogen, 36–4401, 1:500). Alexa Fluor 488, 555 and 647 donkey secondary antibodies were from Jackson ImmunoResearch (1:400). Alexa Fluor 488 conjugated donkey anti-mouse was used to detect murine IgG (1:400, Jackson ImmunoResearch). The anti-uPAR antibody was kindly donated by Francesco Blasi, FIRC Institute of Molecular Oncology Foundation (IFOM), Milan, Italy (*Tjwa et al., 2009*).

## Image acquisition, processing and analysis

Confocal microscopy was performed using confocal microscopes (SP8, Leica). For image analysis, the Fiji software was used (open source, http://fiji.sc/). The Figures were assembled and processed using Adobe Photoshop and Adobe Illustrator. The only adjustments used in the preparation of the Figures were for brightness and contrast. For comparison purposes, different sample images of the same antigen were acquired under constant acquisition settings.

IgG intensity was measured as follows. Each mouse brain sagittal section was acquired under $10 \times$ microscopy. Then, using ImageJ, six regions of interest (ROI) were defined (three in the cortex, one in the hippocampus, one in the cerebellum). The ROI size for the hippocampus and cerebellum were defined to roughly cover 80% of the area of the region. The mean intensities of each ROI were measured. The size and the relative position of the ROIs were kept constant across each comparison. Two sections for each sample were used for quantification.

Quantification of the Ki-67 EC nuclei was carried out as follows. For each confocal stack acquired, the ERG-positive nuclei were first counted using the three-dimensional object counter plugin (*Bolte and Cordelières, 2006*) for ImageJ. ERG staining was then used to create a binary mask of the EC nuclei. Ki-67 stacks were then multiplied by the corresponding binary mask to isolate the EC-specific nuclei. Finally, Ki-67 positive nuclei were counted using the three-dimensional object counter plugin for ImageJ, and the ratio of Ki-67 positive/ERG positive nuclei was calculated. The outline of the ERG-positive nuclei in the binary mask is shown in *Figure 4D* (middle panels, purple) in combination with the original ERG staining (middle panels, green). For visualization purposes, the results of EC filtered Ki-67 are shown in the lower panels of *Figure 4D* (light blue) in combination with PECAM-1 counterstaining.

The uPAR staining intensity was quantified as follows. For each confocal stack acquired, Podocalyxin staining was used to create a binary mask of the ECs and vessels. Then, uPAR stacks were multiplied by the corresponding binary mask to isolate EC vascular-specific staining, preserving the pixel intensity. The pixel mean intensity was then measured on the filtered stacks.

Quantification of tip cells was carried out by two separate operators. The first operator acquired the confocal stacks, and the second operator manually counted tip cells in the brain, tip and tip-like cells in the retina, and filopodia/tip cells in a blind test. Cell counting was through the cell counter plugin of ImageJ.

The radial expansion in the retina was measured as the mean distance covered by the vessels growing from the optic nerve.

## Spatial transcriptomics library preparation and sequencing

*Pdcd10-ko* and *Pdcd10-wt* mice were anesthetized by intraperitoneal injection of Avertin 20 mg/kg (Sigma, T48402), and perfused with ice-cold Hank's balanced salt solution (HBSS) for 3 min. The brains were carefully dissected out from the skulls and snap-frozen using chilled isopentane in a dry-ice bath (according to the suggested protocols for tissue sample collection for Visium; $10\times$ Genomics, Pleasanton, CA, USA). Fresh-frozen brain samples were then embedded in Optimal Cutting Temperature (TissueTek Sakura). Coronal 10-µm-thick cryosections of cerebellum were prepared at $-20°C$ using a cryostat (CryoStar NX50; Thermo Scientific). Quality of the tissue was confirmed with in-situ hybridization using the RNAscope technique, with 3-plex positive controls (ACD, Newark, CA, USA; #320881; for detecting Polr2a, PPIB, UBC mRNA) and 3-plex negative controls (ACD, #320871; for detecting DapB mRNA of *Bacillus subtilis*), according to the protocol described below. Mouse brain samples with good morphology and expressing mRNA of the housekeeping genes throughout the tissue sections were considered as good quality samples.

Fresh-frozen brain samples (n = 2, per genotype) derived from *Pdcd10-wt* and *Pdcd10-ko* mice were chosen for the analysis with the Visium spatial transcriptomics. Here, 10-µm-thick coronal sections were placed on chilled Visium Spatial Gene Expression slides (#PN-2000233, $10\times$ Genomics). Tissue sections were then fixed in chilled methanol and stained with hematoxylin and eosin according to the Visium Spatial Gene Expression user guide (CG000239 Rev A, $10 \times$ Genomics). Brightfield hematoxylin and eosin images were acquired using brightfield and fluorescent microscopy (DMi8; Leica) at $10 \times$ magnification, and exported as TIFF files. After imaging, the tissue sections on the slides were permeabilized for 16 min, which was selected as the optimal time based on tissue

optimization time-course experiments (Visium Tissue Optimization Slides and reagents, #PN-1000193, 10× Genomics) ran prior to the gene expression analysis, following the manufacture instructions.

On-slide cDNA synthesis, tissue removal, probe cleavage, and final library preparation were performed according to the Visium Spatial Gene Expression user guide (CG000239 Rev A, 10 × Genomics). The cDNA and post-library construction quantification were analyzed by running 1 µL samples on High Sensitivity RNA ScreenTape (4200 TapeStation system; Agilent, Santa Clara, CA, USA). Libraries were sequenced (NovaSeq 6000 system; Illumina), with an estimated sequencing depth of 50,000 reads per tissue spot.

## Spatial transcriptomics data analysis

The raw sequencing data and histology images were first processed using the analysis pipeline Space Ranger (v1.0, 10× Genomics) for sample demultiplexing, barcode processing, reads alignment to mouse reference genome (mm10), single-cell 3′ gene counting per spot, and clustering and placing spots in spatial context on the slide image.

Loupe cell browser (v 4.0, 10× Genomics) was used to further explore the clustering results from the Space Ranger pipeline. The number of clusters for each tissue slide was determined based on the anatomical region of the brain (*Goldowitz and Hamre, 1998*) and using the Allen Brain Atlas (http://atlas.brain-map.org/) as reference (*Figure 4—figure supplement 1A*). EC spots were defined as those co-expressing *Cldn5*, *Cdh5*, and *Pecam1* (expression >0.5) (*Figure 4F*, *Figure 4—figure supplement 1B* and *Figure 5—figure supplement 1A*, yellow). Proliferating EC spots were defined as those co-expressing *Cldn5*, *Cdh5*, *Pecam1* (expression >0.5), and *Mki67* (expression >1) (*Figure 4F*, light blue). Proliferating lesion EC spots were defined as those co-expressing *Cldn5*, *Cdh5*, *Pecam1*, *Klf4*, *Klf2*, *Ly6a*, *Thbd* (expression >0.5), and *Mki67* (expression >1) (*Figure 4F*, light green). tip cell EC spots were defined as those co-expressing *Cldn5*, *Cdh5*, *Pecam1*, *Apln*, and *Plaur* (expression >0.5) (*Figure 5—figure supplement 1A and B*, light blue). tip cell lesion EC spots were defined as those co-expressing *Cldn5*, *Cdh5*, *Pecam1*, *Klf4*, *Klf2*, *Ly6a*, *Thbd*, *Apln*, and *Plaur* (expression >0.5) (*Figure 5—figure supplement 1A and B*, light green). Proliferating tip cell EC spots were defined as those co-expressing *Cldn5*, *Cdh5*, *Pecam1*, *Apln*, *Plaur* (expression >0.5), and *Mki67* (expression >1). Proliferating tip cell lesion EC spots were defined as those co-expressing *Cldn5*, *Cdh5*, *Pecam1*, *Klf4*, *Klf2*, *Ly6a*, *Thbd*, *Apln*, *Plaur* (expression >0.5), and *Mki67* (expression >1). The Fisher's exact test was used to analyze whether the proportions of proliferating spots are different between endothelial and lesion endothelial spots, or between tip cell endothelial or tip cell lesion endothelial spots in *Pdcd10-ko* samples. The total spots count from both *Pdcd10-ko* samples were merged and used for the Fisher's exact test.

The R package Seurat (v.3.1) was used to perform integrated analysis of the spatial transcriptomic data. Similar to joined scRNA-seq data analysis, data from all of the slices were normalized and merged. Then merged data were used for dimensional reduction and clustering. The gene expression levels in all of the spots were overlaid on tissue staining images to visualize the spatial location of the selected spots.

## In-situ hybridization using the RNAscope technique

In-situ hybridization using the RNAscope technique was performed according to the manufacturer instruction (Advanced Cell Diagnostics [ACD]), using multiplex fluorescent Reagent Kit v2. Briefly, fresh-frozen brain samples derived from *Pdcd10-wt* and *Pdcd10-ko* mice were prepared as described above. Then 10-µm-thick coronal cryosections were pretreated with $H_2O_2$, followed by a 15 min incubation with protease IV. The probes *Cldn5* (ACD, #491611-C2) and *Bst1* (ACD, #559841-C3) and the 3-plex negative and positive controls were hybridized on the tissue sections for 2 hr at 40˚C (HybEZ oven; ACD), followed by signal amplification with the reagents included in Reagent Kit v2. The signal was detected with TSA plus fluorescein cyanine 3 (for *Cldn5*) and cyanine 5 (for *Bst1*; all dyes from Akoya Bioscience Inc, Marlborough, MA, USA), and tissue sections were counterstained with DAPI (ACD). Fluorescent tilescan images that visualized the signals for the mRNA expression were acquired using a fluorescent microscope (DMi8; Leica) at 20 × magnification. Representative close-up images were then acquired with the confocal microscope at 20 × and 40 × magnification.

For comparison purposes, different sample images of the same probe combinations were acquired under constant acquisition settings.

## Acknowledgements

We thank Christopher Berrie for editorial assistance. We thank and Ralf H Adams and Francesco Blasi for generously sharing their genetically modified mouse strain and uPAR antibody. We also thank Christer Betsholtz for sharing his genetically modified mouse strain and for critically reviewing the manuscript.

## Additional information

### Competing interests

Elisabetta Dejana: Reviewing editor, *eLife*. The other authors declare that no competing interests exist.

### Funding

| Funder | Grant reference number | Author |
|---|---|---|
| Vetenskapsrådet | 2013-9279 | Lei Liu Conze<br>Suvi Jauhiainen<br>Veronica Sundell<br>Sara Isabel Cunha<br>Johan Brännström<br>Maria Ascención Globisch<br>Peetra Magnusson<br>Elisabetta Dejana |
| Knut och Alice Wallenbergs Stiftelse | 2015-0030 | Lei Liu Conze<br>Suvi Jauhiainen<br>Veronica Sundell<br>Sara Isabel Cunha<br>Johan Brännström<br>Maria Ascención Globisch<br>Peetra Magnusson<br>Elisabetta Dejana |
| Associazione Italiana per la Ricerca sul Cancro | AIRC IG 23223 | Fabrizio Orsenigo<br>Monica Corada<br>Matteo Malinverno<br>Claudio Maderna<br>Maria Grazia Lampugnani<br>Elisabetta Dejana |
| Associazione Italiana per la Ricerca sul Cancro | 5x1000 MYNERVA AIRC 21267 | Monica Corada<br>Francesca Lazzaroni<br>Elisabetta Dejana |
| Agenzia Italiana del Farmaco, Ministero della Salute | AIFA-2016-02364593 | Fabrizio Orsenigo<br>Monica Corada<br>Francesca Lazzaroni<br>Matteo Malinverno<br>Claudio Maderna<br>Maria Grazia Lampugnani<br>Elisabetta Dejana |
| H2020 European Research Council | 74292 | Fabrizio Orsenigo<br>Monica Corada<br>Francesca Lazzaroni<br>Matteo Malinverno<br>Claudio Maderna<br>Maria Grazia Lampugnani<br>Elisabetta Dejana |
| H2020 Marie Skłodowska-Curie Actions | 675619 | Elisabetta Dejana |
| Fondazione Telethon | GGP19202 | Fabrizio Orsenigo<br>Monica Corada |

Matteo Malinverno
Claudio Maderna
Maria Grazia Lampugnani
Elisabetta Dejana

The funders had no role in study design, data collection and interpretation, or the decision to submit the work for publication.

## Author contributions

Fabrizio Orsenigo, Conceptualization, Data curation, Validation, Investigation, Visualization, Methodology, Writing - review and editing; Lei Liu Conze, Conceptualization, Data curation, Software, Formal analysis, Visualization, Methodology, Writing - review and editing; Suvi Jauhiainen, Validation, Investigation, Methodology; Monica Corada, Francesca Lazzaroni, Matteo Malinverno, Resources, Writing - review and editing; Veronica Sundell, Sara Isabel Cunha, Johan Brännström, Claudio Maderna, Resources; Maria Ascención Globisch, Resources, Methodology; Maria Grazia Lampugnani, Conceptualization, Data curation, Supervision, Writing - original draft; Peetra Ulrica Magnusson, Conceptualization, Supervision, Project administration, Writing - review and editing; Elisabetta Dejana, Conceptualization, Supervision, Funding acquisition, Writing - original draft, Project administration

## Author ORCIDs

Fabrizio Orsenigo (iD) https://orcid.org/0000-0001-9135-8478
Monica Corada (iD) http://orcid.org/0000-0001-8220-0871
Francesca Lazzaroni (iD) http://orcid.org/0000-0001-5767-7846
Matteo Malinverno (iD) https://orcid.org/0000-0001-8242-7937
Claudio Maderna (iD) https://orcid.org/0000-0002-1404-5408
Maria Grazia Lampugnani (iD) http://orcid.org/0000-0002-4802-7064
Peetra Ulrica Magnusson (iD) https://orcid.org/0000-0003-1142-854X
Elisabetta Dejana (iD) https://orcid.org/0000-0002-0007-0426

## Ethics

Animal experimentation: The experimental animal protocols were approved by the Uppsala Ethical Committee on Animal Research (permit number C145/15) and the Italian Ministry of Health. Animal procedures were performed in accordance with the Institutional Animal Care and Use Committee (IACUC) and in compliance with the guidelines established in the Principles of Laboratory Animal Care (directive 86/609/EEC).

## Decision letter and Author response

Decision letter https://doi.org/10.7554/eLife.61413.sa1
Author response https://doi.org/10.7554/eLife.61413.sa2

## Additional files

### Supplementary files

• Supplementary file 1. Marker genes used in *Figure 1H*, with supporting references.

• Supplementary file 2. Known lesion marker genes, showing average logFC in each cluster and bibliographic references. The table refers to *Figure 2C* and shows, for the genes included in the heatmap: (1) gene symbol, with basic gene informations; (2) the average logFC observed in each cluster (*Pdcd10-ko* vs. *Pdcd10-wt*); (3) a selected list of bibliographic references supporting the choice of each marker.

• Supplementary file 3. EnrichR results from RareDisease GeneRif dataset. The table refers to *Figure 2D* and show the results of the over-representation analysis. The table is showing, for cerebral cavernous malformation term: (1) the cluster that showed significant enrichment; (2) the DEGs contributing to the enrichment, with basic gene informations; (3) the average logFC observed in each cluster (*Pdcd10-ko* vs. *Pdcd10-wt*).

• Supplementary file 4. Over-representation analysis results. The table refers to *Figure 2E* and show the results of the over-representation analysis. Hallmarks are divided in categories, as in the figure. For each hallmark is shown: (1) the cluster that showed significant enrichment; (2) the DEGs contributing to the enrichment, with basic gene informations; (3) the average logFC observed in each cluster (*Pdcd10-ko* vs. *Pdcd10-wt*).

• Supplementary file 5. References supporting the annotation of unique marker genes of cluster eight set in alphabetical order and summarized in *Figure 3A*.

• Supplementary file 6. References supporting the annotation of DEGs (*Pdcd10-ko vs. Pdcd10-wt*) of cluster eight set in alphabetical order and summarized in *Figure 3D*.

• Supplementary file 7. Canonical marker genes of clusters. The differentially expressed genes with avg_logFC > 0 in both genotypes and minimump_p_val < 0.05 are considered as canonical markers of clusters.

• Supplementary file 8. Differentially expressed genes between *Pdcd10 (Ccm3)* ko and *Pdcd10 (Ccm3)* wt in each cluster. The genes are considered differentially expressed if p_val_adj < 0.05.

• Supplementary file 9. Details of statistical analysis.

• Transparent reporting form

## Data availability

Sequencing data have been deposited in GEO under accession code GSE155788.

The following dataset was generated:

| Author(s) | Year | Dataset title | Dataset URL | Database and Identifier |
|---|---|---|---|---|
| Orsenigo F, Conze LL, Jauhiainen S, Corada M, Lazzaroni F, Malinverno M, Sundell V, Cunha SI, Brännström J, Globish M, Maderna C, Lampugnani MG, Magnusson PU, Dejana E | 2020 | Mapping endothelial-cell diversity in cerebral cavernous malformations at single cell resolution | https://www.ncbi.nlm.nih.gov/geo/query/acc.cgi?acc=GSE155788 | NCBI Gene Expression Omnibus, GSE155788 |

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
