## [Decision Letter]

**Acceptance summary:**

The manuscript by Orsenigo and colleagues summarizes the results of a single cell RNA sequencing study focussing on the molecular nature of those cells that actually trigger cavernoma formation in a murine CCM3 endothelial-specific knock out model. This is the first report of such an analysis at single cell level of ECs in CCM lesions. This study is very timely since it has been shown only recently that the CCM pathology is caused by clones of CCM mutant ECs that can recruit neighboring wild-type cells into cavernoma. The substantial and impressive amount of data presented and the conclusions discussed in this manuscript are very interesting and relevant to the field of CCM research and beyond.

**Decision letter after peer review:**

Thank you for submitting your article "Mapping endothelial-cell diversity in CCMs at single cell resolution" for consideration by *eLife*. Your article has been reviewed by three3 peer reviewers, including Salim Abdelilah-Seyfried as the Reviewing Editor and Reviewer #1, and the evaluation has been overseen by Didier Stainier as the Senior Editor. The following individual involved in review of your submission has agreed to reveal their identity: Luisa Iruela-Arispe (Reviewer #2).

The reviewers have discussed the reviews with one another and the Reviewing Editor has drafted this decision to help you prepare a revised submission.

Summary:

The manuscript by Orsenigo and colleagues summarizes the results of a single cell RNA sequencing study focussing on the molecular nature of those cells that actually trigger cavernoma formation in a murine CCM3 endothelial-specific knock out model. This is the first report of such an analysis at single cell level of ECs in CCM lesions. This study is very timely since it has been shown only recently that the CCM pathology is caused by clones of CCM mutant ECs that can recruit neighboring wild-type cells into cavernoma.

The biology uncovered is extremely interesting, as it reveals that these cells might not be able to interpret flow signals adequately as they showed significant upregulation of Klf2 and 4. The link between interpretation of flow signals is further supported by the fact that in these mice arterial ECs are protected from *Ccm3* deletion. They finally identify *Bst1* as an important physiological marker of vein-resident progenitor cells that appears to "supply" cells in the cavernous malformations.

The substantial and impressive amount of data presented and the conclusions discussed in this manuscript are very interesting and relevant to the field of CCM research and beyond. The study is comprehensive, beautifully presented, well-controlled and the conclusions put forward with one two exceptions (see below) are supported by solid preliminary data. Here, some textual changes are required.

Essential revisions:

1) The authors present a major finding that mainly venous capillary and vessel-resident progenitor ECs are prone to cavernoma malformation. This finding provides a molecular correlate to the medical characterization of this pathology as a venous capillary malformation. Based on this finding, the authors also conclude that arterial cells are apparently "protected" from CCM transformation. However, the term "protected" implies some molecular mechanism of protection. Is it possible that there are more molecular changes among venous cells, when compared with arterial cells, simply because CCM transformations occur by definition mostly in venous beds?

The identification of "protective" genes in the C8 (arterial cluster) in relation to CCMs might be a stretch. These are DEG genes, it is unclear whether they convey a protective effect (functional). It is clear that the arteries are protected, but is this due to other factors (flow, distinct heterotypic cell interactions, etc) or due to these genes. The concern here is labeling protective. The data does not show that – at least not at this point. The use of the literature for this support is interesting, but does it really relate to CCM3? Thus, this reviewer does takes an objection with the solid conclusion that "these arterial ECs express genes that can counteract the effects of *Ccm3* deletion, and thus prevent lesion formation" – This has not been shown. In fact, in the Discussion the authors do say: "It is possible that the unique resistance of these cells is induced and maintained by up-regulation of a relatively large set of "defensive" genes that are modulated directly or indirectly by *Ccm3*".

2) The potential mechanisms protecting arterial ECs against phenotypic transformation are interesting but remain hypothetical at this stage. The categorization of genes as protective or harmful may be somehow superficial and heavily context-dependent. Both points should be reflected in the Discussion.

Insights or discussion regarding the specificity of *Bst1* and *Peg3* as opposed to *Procr* and *Cd200* as markers of resident endothelial progenitors with pathogenic potential would be interesting.

Unfortunately, the identity of resident endothelial progenitors remains elusive, as these cells do not seem to represent a distinct population in the scRNA-seq results. Instead, the authors propose that progenitor cells are heterogenous and similar to other ECs in the surrounding vessel bed. In this context, it might be worthwhile to consider (and discuss) other possibilities. For example, could "progenitor" status represent a transient response to microenvironmental signals rather than a hard-wired, stable identity.

3) The analysis of the single-cell sequencing data is comprehensive, but becomes sometimes difficult to follow especially due to the large number of clusters that were identified. It is, for example, a bit surprising that the authors propose 3 different tip cell subpopulations. Apart from a number of differentially expressed genes, can this be attributed to regional, functional or other differences? Are the results reproducible using a resolution that would yield less clusters by grouping cells of a rather similar kind?

---

## [Author Response]

Essential revisions:1) The authors present a major finding that mainly venous capillary and vessel-resident progenitor ECs are prone to cavernoma malformation. This finding provides a molecular correlate to the medical characterization of this pathology as a venous capillary malformation. Based on this finding, the authors also conclude that arterial cells are apparently "protected" from CCM transformation. However, the term "protected" implies some molecular mechanism of protection. Is it possible that there are more molecular changes among venous cells, when compared with arterial cells, simply because CCM transformations occur by definition mostly in venous beds?The identification of "protective" genes in the C8 (arterial cluster) in relation to CCMs might be a stretch. These are DEG genes, it is unclear whether they convey a protective effect (functional). It is clear that the arteries are protected, but is this due to other factors (flow, distinct heterotypic cell interactions, etc) or due to these genes. The concern here is labeling protective. The data does not show that – at least not at this point. The use of the literature for this support is interesting, but does it really relate to CCM3? Thus, this reviewer does takes an objection with the solid conclusion that "these arterial ECs express genes that can counteract the effects of Ccm3 deletion, and thus prevent lesion formation" – This has not been shown. In fact, in the Discussion the authors do say: "It is possible that the unique resistance of these cells is induced and maintained by up-regulation of a relatively large set of "defensive" genes that are modulated directly or indirectly by Ccm3"

We agree with the reviewer that the experimental evidence of the ‘protective or harmful’ role of the unique genes expressed in arterial C8 is missing. Therefore, we have extensively reshaped the section around arterial C8 (subsection “Arterial differentiation prevents ECs from forming CCM lesions”) and we have strongly lightened the conclusion from this paragraph. In Figure 3, panel B is now presented as a speculative interpretation of the data shown in Figure 3A. We hypothesize ‘putative preventive or promotive functions’ for these genes considering, on one hand, their activity as reported in the literature (Supplementary file 6), and on the other, the known deregulation of functions and signalling after *Ccm3* deletion. Suggestively, many of these unique genes positively or negatively regulate functions that are altered in the *Ccm3*-ko.

We have also tried to clarify some concepts that were obscure or implicit in the previous version. This includes the list of genes in Figure 3A referred to as ‘Unique markers genes’, which are genes that are highly expressed uniquely in C8 both in *Ccm3*-wt and *Ccm3*-ko cells. Therefore, the genes in Figure 3A represent a hallmark of this cluster.

The unique marker genes of C8 might reasonably be part of the transcriptional response of arterial ECs to specific environmental stimuli, such as flow, heterotypic interactions, extracellular matrix, etc., that builds arterial refractoriness to develop the mutated phenotype.

There are only seven unique differentially expressed genes in the C8 *Ccm3*-ko in comparison to *Ccm3*-wt, and these are discussed in the same section. Also for these genes, the annotation (using https://pubmed.ncbi.nlm.nih.gov/ and http://www.informatics.jax.org/) suggests preventive functions. As these genes are up-regulated specifically in the *Ccm3*-ko of C8 in comparison to the respective *Ccm3*-wt, they might also be part of the specific responses of arterial ECs to *Ccm3* deletion, and might contribute to the maintenance of the ‘normal’ phenotype of the *Ccm3*-ko ECs of arterial C8.

2) The potential mechanisms protecting arterial ECs against phenotypic transformation are interesting but remain hypothetical at this stage. The categorization of genes as protective or harmful may be somehow superficial and heavily context-dependent. Both points should be reflected in the Discussion.

We agree with the reviewer that our findings here are still at a hypothetical stage. Therefore, we have extensively modified and lightened our comments and interpretation of the mechanisms of protection of the arterial ECs from expressing the *Ccm3*-ko phenotype. We describe these changes in the answer to point 1 (above).

Insights or discussion regarding the specificity of Bst1 and Peg3 as opposed to Procr and Cd200 as markers of resident endothelial progenitors with pathogenic potential would be interesting.Unfortunately, the identity of resident endothelial progenitors remains elusive, as these cells do not seem to represent a distinct population in the scRNA-seq results. Instead, the authors propose that progenitor cells are heterogenous and similar to other ECs in the surrounding vessel bed. In this context, it might be worthwhile to consider (and discuss) other possibilities. For example, could "progenitor" status represent a transient response to microenvironmental signals rather than a hard-wired, stable identity.

We agree that a discussion regarding the specificity of markers of resident endothelial progenitors with pathogenic potential would be of interest. The issues of specific functions for progenitors expressing different progenitor markers, as well as that of the modulation of progenitor state by the microenvironment, are really challenging and deserve future experimental investigation.

To comment on these aspects:

– We now discuss some findings reported in the literature on the role of different types of progenitors under different experimental conditions that mimic pathologies (Discussion, fourth paragraph).

– We now discuss the concept that the specific microenvironment might shape the transcriptome of the progenitors to induce a sort of ‘mimicry’ with the ECs of the different vessels (Discussion, first paragraph). We have also introduced the concept that the ‘progenitor’ state might be a transient, reversible condition, instead of a stable, fate-committed identity.

3) The analysis of the single-cell sequencing data is comprehensive, but becomes sometimes difficult to follow especially due to the large number of clusters that were identified. It is, for example, a bit surprising that the authors propose 3 different tip cell subpopulations. Apart from a number of differentially expressed genes, can this be attributed to regional, functional or other differences? Are the results reproducible using a resolution that would yield less clusters by grouping cells of a rather similar kind?

The choice of resolution for definition of the clusters was mentioned briefly in the Materials and methods of our manuscript. We agree with the reviewer that better clarification will be useful to help readers to follow our comprehensive analysis. Therefore, we have now added a specific section in the Materials and methods (subsection “Selection and evaluation of clustering resolution”) and two new/updated figures describing the criteria applied to select the resolution level.

When we performed clustering using Seurat, we varied the resolution parameter from 0.01 to 1 (see revised Figure 1—figure supplement 3B). Seurat applies a graph-based clustering approach, and a resolution parameter controls the partitioning of the cellular distance matrix into clusters; i.e. higher resolution results in more clusters. To explore and examine the impact of varying clustering resolution, we then used the cluster package to produce a clustering tree to visualize how the cells are positioned as the number of clusters in our dataset increases. The clustering tree covering resolutions 0.01 to 1 (revised Figure 1—figure supplement 3B) shows that most of the branching occurs in the branch starting with cluster 0, which consistently has subbranches split off to form new clusters as the resolution increases. There are two regions of stability in this tree: at resolution 0.09–0.1 and at resolution 0.4–0.5, where no additional sub-branching occurred in between, and the number of clusters stayed the same. At resolution 0.5, we begin to see lower in-proportion edges and new clusters that form from multiple parent clusters. These are signs of over-clustering; i.e. the algorithm is forced to produce more clusters than are likely to be truly present in this dataset.

After we assigned the prevalent cell type identities to the clusters at resolution 0.4, we overlaid the cluster identity information onto the cluster tree (new Figure 1—figure supplement 4). The aim was to check whether each cluster represent a true sub-endothelial cell population, and whether we have over-clustered our dataset at resolution 0.4. The new Figure 1—figure supplement 4 show that:

A) two of the four contaminant cell clusters (C10 and C15) are branched at the lowest resolution tested (0.01, black line).

B1) the two major tip cell clusters (C1 and C6) derive from two distinct lineages that are branched already at resolution 0.03 (red line). This suggests that C1 and C6 are not the result of over-clustering, but rather two distinct tip cell sub-populations. This was further confirmed by our trajectory analysis (see Figure 6A), where clusters 1 and 6 clearly differentiated into two separate lineages. As suggested by the reviewer, this might be attributed to regional or functional differences, but we believe that such investigation is beyond the scope of this manuscript.

B2) arterial and venous clusters (C8 and C9, respectively) also derive from two distinct lineages that are branched at 0.03 resolution (red line), which reflects the great differences between these two ECs sub-populations.

C) the remaining two contaminant cell clusters (C11 and C16) were instead separated from ECs at 0.05 and 0.2 resolution, respectively (dashed black lines).

D) the mitotic/venous capillary and the venous/venous capillary clusters (C7 and C9, respectively) branched out, from the same pool of cells at resolution 0.4 (dashed green line). This late separation suggests a high similarity between the cells in these two clusters, which turned out to be the most affected clusters in *Ccm3* ko.

In conclusion here, we have chosen the lowest resolution that combined good isolation of contaminant cells (i.e. ≥0.2), biologically meaningful clustering of ECs, and the need to avoid over-clustering. Considering all of the observations listed above, 0.4 turned out to be the optimal resolution for the purpose of this study.